# Revisiting Deep Feature Reconstruction for Logical and Structural Industrial Anomaly Detection

**Sukanya Patra**                                          *sukanya.patra@umons.ac.be*
*University of Mons*

**Souhaib Ben Taieb**                                      *souhaib.bentaieb@mbzuai.ac.ae*
*Mohamed bin Zayed University of Artificial Intelligence*
*University of Mons*

Reviewed on OpenReview: *https://openreview.net/forum?id=kdTC4ktHPD*

## Abstract

Industrial anomaly detection is crucial for quality control and predictive maintenance, but it presents challenges due to limited training data, diverse anomaly types, and external factors that alter object appearances. Existing methods commonly detect structural anomalies, such as dents and scratches, by leveraging multi-scale features from image patches extracted through deep pre-trained networks. However, significant memory and computational demands often limit their practical application. Additionally, detecting logical anomalies—such as images with missing or excess elements—requires an understanding of spatial relationships that traditional patch-based methods fail to capture. In this work, we address these limitations by focusing on Deep Feature Reconstruction (DFR), a memory- and compute-efficient approach for detecting structural anomalies. We further enhance DFR into a unified framework, called ULSAD, which is capable of detecting both structural and logical anomalies. Specifically, we refine the DFR training objective to improve performance in structural anomaly detection, while introducing an attention-based loss mechanism using a global autoencoder-like network to handle logical anomaly detection. Our empirical evaluation across five benchmark datasets demonstrates the performance of ULSAD in detecting and localizing both structural and logical anomalies, outperforming eight state-of-the-art methods. An extensive ablation study further highlights the contribution of each component to the overall performance improvement. Our code is available at https://github.com/sukanyapatra1997/ULSAD-2024.git.

## 1 Introduction

Anomaly detection (AD) is a widely studied problem in many fields that is used to identify rare events or unusual patterns (Salehi et al., 2022). It enables the detection of abnormalities, potential threats, or critical system failures across diverse applications such as predictive maintenance (PdM) (Tang et al., 2020; Choi et al., 2022), fraud detection (Ahmed et al., 2016; Hilal et al., 2022), and medicine (Tibshirani & Hastie, 2007; Fernando et al., 2021). Despite its importance and widespread applicability, it remains a challenging task as characterising anomalous behaviours is difficult and the anomalous samples are not known a priori (Ruff et al., 2021). Therefore, AD is often defined as an unsupervised representation learning problem (Pang et al., 2020; Reiss et al., 2022) where the training data contains predominantly normal samples. The aim is to learn the normal behaviour using the samples in the training set and identify anomalies as deviations from this normal behaviour. This setting is also known as one-class classification (Ruff et al., 2018).

Our study focuses on Industrial Anomaly Detection (IAD) (Bergmann et al., 2019), with an emphasis on the detection of anomalies in images from industrial manufacturing processes. Image-based IAD methods assign an anomaly score to each image. Further, this study looks into anomaly localization where each pixel of an image is assigned an anomaly score. It enables fine-grained localization of the anomalous regions in the image. Over the years, it has attracted attention from both industry and academia as AD can be used for various tasks like quality control or predictive maintenance, which are of primal interest to industries. Despite the

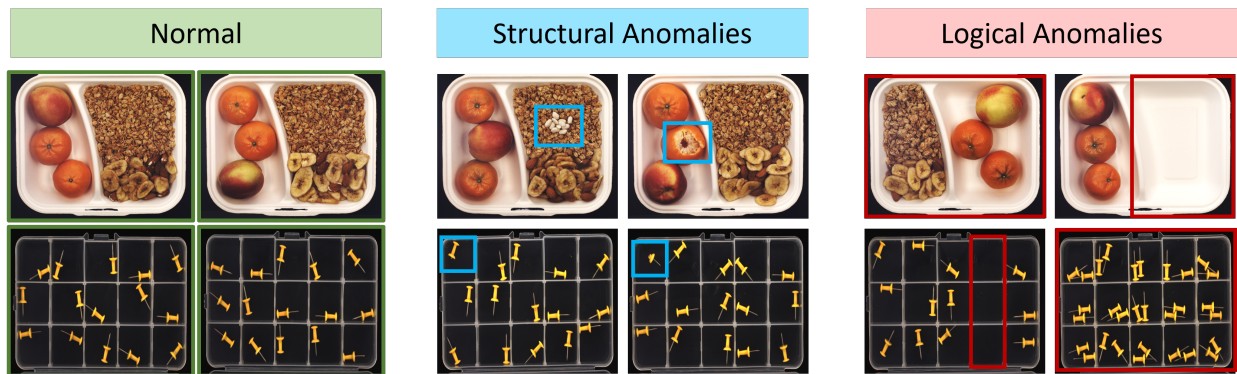

Figure 1: Types of anomalies for the categories "breakfast box" and "pushpin" in MVTecLOCO dataset. Two normal samples (Left) along with structural (Middle) and logical anomalies (Right) where the anomalous regions are highlighted in blue and red, respectively.

necessity, addressing IAD is challenging because: (i) evolving processes result in different manifestations of anomaly (Gao et al., 2023), and (ii) object appearances vary due to external factors such as background, lighting conditions and orientation (Jezek et al., 2021). Furthermore, the anomalies in IAD can be broadly categorized into: (i) **structural anomalies** where subtle localized structural defects can be observed in the images (Bergmann et al., 2019), and (ii) **logical anomalies** where violations of logical constraints result in anomalies (Bergmann et al., 2022). Figure 1 shows examples of both structural and logical anomalies in industrial images.

To detect **structural anomalies**, state-of-the-art methods divides the image into smaller patches and leverage multi-scale features of the image patches obtained using deep convolutional neural networks (Salehi et al., 2021). PatchCore (Roth et al., 2022) and CFA(Lee et al., 2022) achieved state-of-the-art (SOTA) performance by storing the extracted features in a memory bank during the training phase and comparing the features of the image with their closest neighbour from the memory bank during inference. However, such approaches require considerable storage to accumulate the extracted features, which can be challenging for large-scale datasets. The first alternative is knowledge distillation-based approaches (Bergmann et al., 2020; 2022), where a student network is trained to mimic the teacher for normal samples. During inference, anomalies are identified based on the discrepancy between the student and teacher output. A key requirement of these distillation-based approaches is that the student network must be less expressive than the teacher to prevent it from mimicking the teacher on anomalous samples. Thus, regularization methods, such as penalty based on an external dataset or hard-mining loss (Batzner et al., 2024) are applied, that slows down the training and increases the requirement of computing resources. Moreover, excessive regularization can prevent learning representations for normal images. The second alternative is to model the features of normal images using a multivariate Gaussian distribution (Defard et al., 2021) or learn to reconstruct the features using a deep feature reconstruction (DFR) network (Yang et al., 2020).

Besides structural anomalies, **logical anomalies** occur when elements in the images are missing, misplaced, in surplus or violate geometrical constraints (Bergmann et al., 2022). Methods relying on multi-scale features of image patches would fail as they would still be considered normal. It is the combination of objects in the image that makes the image anomalous. Thus, to detect such logical anomalies, it is necessary to look beyond image patches and develop a global understanding of the spatial relationships within normal images. Distillation-based methods, which are predominantly used for the detection of logical anomalies, rely on an additional network to learn the spatial relationships between items in the normal image (Batzner et al., 2024).

In this paper, we focus on DFR, the benefits of which are four-fold. First, it does not need large memory for storing the features, unlike PatchCore (Roth et al., 2022). Second, unlike PaDiM (Defard et al., 2021), it does not make any assumption about the distribution of features. Third, learning to reconstruct features in the latent space of a pre-trained network is less impacted by the curse of dimensionality than learning to reconstruct images which are high-dimensional. Fourth, deep networks trained to reconstruct normal images using the per-pixel distance suffer from the loss of sharp edges of the objects or textures in the background.

As a consequence, AD performance deteriorates due to an increase in false positives, i.e., the number of normal samples falsely labelled anomalies. On the contrary, computing the distance features maps and their corresponding reconstructions during training is less likely to result in such errors (Assran et al., 2023).

We revisit DFR to develop a unified framework for the detection of both structural and logical anomalies. First, we modify the training objective by considering a combination of $\ell_2$ and cosine distances between each feature and the corresponding reconstruction. The incorporation of the cosine distance addresses the curse of dimensionality, where high-dimensional features become orthogonal to each other in Euclidean space and the notion of distance disappears (Aggarwal et al., 2001). Second, to simultaneously allow for the detection of logical anomalies, we introduce an attention-based loss using a global autoencoder-like network. We empirically demonstrate that with our proposed changes, not only do the detection and localization capabilities of DFR improve for structural anomalies, but also it delivers competitive results on the detection of logical anomalies. Our contributions can be summarized as:

- Building on DFR, we propose a **U**nified framework for **L**ogical and **S**tructural **A**nomaly **D**etection referred as `ULSAD`, a framework for detection and localization of both structural and logical anomalies.

- To detect structural anomalies, we consider both magnitude and angular differences between the extracted and reconstructed feature vectors.

- To detect logical anomalies, we propose a novel attention-based loss for learning the logical constraints.

- We demonstrate the effectiveness of `ULSAD` by comparing it with 8 SOTA methods across 5 widely adopted IAD benchmark datasets.

- Through extensive ablation study, we show the effect of each component of `ULSAD` on the overall performance of the end-to-end architecture.

## 2 Related Work

Several methods have been proposed over the years for addressing Industrial AD (Bergmann et al., 2019; 2022; Jezek et al., 2021). They can be broadly categorized into feature-embedding based and reconstruction-based methods. We briefly highlight some relevant works in each of the category. For an extended discussion on the prior works we refer the readers to the survey by Liu et al. (2024).

**Feature Embedding-based methods**. There are mainly three different types of IAD methods which utilize feature embeddings from a pre-trained deep neural network: *memory bank* (Defard et al., 2021; Roth et al., 2022; Lee et al., 2022), *student-teacher* (Zhang et al., 2023; Batzner et al., 2024), and *density-based* (Gudovskiy et al., 2021; Yu et al., 2021). The main idea of *memory bank* methods is to extract features of normal images and store them in a memory bank during the training phase. During the testing phase, the feature of a test image is used as a query to match the stored normal features. There are two main constraints in these methods: *how to learn useful features* and *how to reduce the size of the memory bank*. While PatchCore (Roth et al., 2022) introduces a coreset selection algorithm, CFA (Lee et al., 2022) clusters the features in the memory bank to reduce the size of the memory bank. Nonetheless, the performance of the *memory bank* methods heavily depends on the completeness of the memory bank, which requires a large number of normal images. Moreover, the memory size is often related to the number of training images, which makes these methods not suitable for large datasets or very high-dimensional images. In the *student-teacher* approach, the student network learns to extract features of the normal images, similar to the teacher model. For anomalous images, the features extracted by the student network are different from the teacher network. Batzner et al. (2024) propose to use an autoencoder model in addition to the student network to identify logical anomalies. For leveraging the multiscale feature from the teacher network to detect anomalies at various scales, Deng & Li (2022) propose Reverse Distillation. Zhang et al. (2023) extended it by proposing to utilize two student networks to deal with structural and logical anomalies. Yang et al. (2020) propose to learn a deep neural network for learning to reconstruct the features of the normal images extracted using the pre-trained backbone. For *density-based* methods, first, a model is trained to learn the distribution of the features obtained from normal samples. Then, during inference, anomalies are detected based on the likelihood of features extracted from the test images. PaDiM (Defard et al., 2021) uses

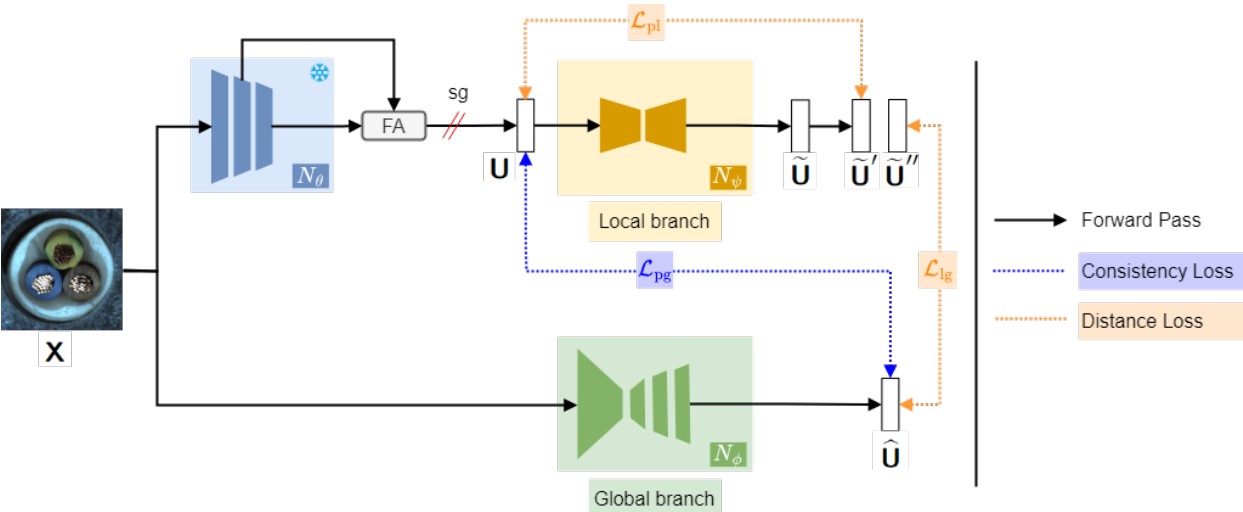

Figure 2: Overview of the end-to-end architecture of `ULSAD`.

a multivariate Gaussian to estimate the density of the features corresponding to the samples from the normal class while FastFlow (Yu et al., 2021) and CFLOW (Gudovskiy et al., 2021) utilize normalizing flows.

**Reconstruction-based methods**. Reconstruction-based methods assume that encoder-decoder models trained on normal samples will exhibit poor performance for anomalous samples. However, relying solely on the reconstruction objective can result in the model collapsing to an identity mapping. To address this, structural assumptions are made regarding the data generation process. One such assumption is the *Manifold Assumption*, which posits that the observed data resides in a lower-dimensional manifold within the data space. Methods leveraging this assumption impose a bottleneck by restricting the encoded space to a lower dimensionality than the actual data space. Common deep reconstruction models used include AE or VAE-based approaches. Advanced strategies encompass techniques like reconstruction by memorised normality (Gong et al., 2019), model architecture adaptation (Lai et al., 2019) and partial/conditional reconstruction (Yan et al., 2021; Nguyen et al., 2019). Generative models like GANs are also widely employed for anomaly detection, as the discriminator inherently calculates the reconstruction loss for samples (Zenati et al., 2018). Variants of GANs, such as denoising GANs (Sabokrou et al., 2018) and class-conditional GANs (Perera et al., 2019), improve anomaly detection performance by increasing the challenge of reconstruction. Some methods utilize the reconstructed data from GANs in downstream tasks to enhance the amplification of reconstruction errors for anomaly detection (Zhou et al., 2020). Lastly, DRÆM (Zavrtanik et al., 2021) trains an additional discriminative network alongside a reconstruction network to improve the AD performance.

In this paper, we focus on feature embedding-based methods motivated by their effectiveness in the current SOTA methods. Specifically, we build on DFR (Yang et al., 2020), which has several benefits. First, it is memory-efficient as it does not rely on a memory bank of extracted features, unlike PatchCore (Roth et al., 2022). Second, unlike PaDim (Defard et al., 2021), it does not make any assumption about the distribution of the extracted features. Third, it is computationally efficient and less impacted by the curse of dimensionality as it operates in the lower-dimensional latent space of a deep neural network. Last, by avoiding the use of per-pixel distance in its reconstruction objective, it is less prone to false positives (Assran et al., 2023).

## 3 The `ULSAD` Framework for Anomaly Detection

We propose `ULSAD`, a framework for simultaneously detection and localization of anomalies in images as shown in Figure 2. Firstly, we utilize a feature extractor network for extracting low-dimensional features from high-dimensional images, which we discuss in Section 3.1. Then, for the detection of both structural and logical anomalies, we rely on a dual-branch architecture. The local branch detects structural anomalies with the help of a feature reconstruction network applied to the features corresponding to patches in the image. We elaborate on this in Section 3.2. Conversely, the global branch, as discussed in Section 3.3, detects logical

anomalies using an autoencoder-like network, which takes as input the image. Lastly, we provide an overview of the `ULSAD` algorithm in Section 3.4 followed by a discussion on the inference process in Section 3.5.

We consider a dataset $\mathcal{D} = \{(\mathbf{X}_i, y_i)\}_{i=1}^{n}$ with $n$ samples where $\mathbf{X}_i \in \mathcal{X}$ is an image and $y_i \in \{0, 1\}$ is the corresponding label. We refer to the normal class with the label 0 and the anomalous class with the label 1. The samples belonging to the anomalous class can contain either logical or structural anomalies. We denote the train, validation and test partitions of $\mathcal{D}$ as $\mathcal{D}_{\text{train}}$, $\mathcal{D}_{\text{valid}}$ and $\mathcal{D}_{\text{test}}$, respectively. The training and validation sets contains only normal samples, i.e., $y = 0$. For the sake of simplicity, we refer to the training set as $\mathcal{D}_N = \{\mathbf{X} \,|\, (\mathbf{X}, 0) \in \mathcal{D}_{\text{train}}\}$. The test set $\mathcal{D}_{\text{test}}$ includes both normal and anomalous samples.

### 3.1 Feature Extractor

High-dimensional images pose a significant challenge for AD (Reiss et al., 2022). Recent studies have shown that deep convolutional neural networks (CNNs) trained on ImageNet (Russakovsky et al., 2015) capture discriminative features for several downstream tasks. Typically, AD methods (Salehi et al., 2021; Defard et al., 2021; Yoon et al., 2023) leverage such pre-trained networks to extract features maps corresponding to partially overlapping regions or patches in the images. Learning to detect anomalies using the lower-dimensional features is beneficial as it results in reduced computational complexity. A key factor determining the efficiency of such methods is the size of the image patches being used, as anomalies can occur at any scale. To overcome this challenge, feature maps are extracted from multiple layers of the CNNs and fused together (Salehi et al., 2021; Roth et al., 2022; Yang et al., 2020). Each element in a feature map obtained from different layers of a convolutional network corresponds to a patch of a different size in the image depending on its receptive field. Thus, combining feature maps from multiple layers results in multi-scale representation of the image patches, which we refer to as **patch features**.

Similar to DFR, we extract low-dimensional feature maps by combining features from multiple layers of a feature extractor which is a pre-trained CNN $N_\theta$ parameterized by $\theta$. In this paper, we consider ResNet-like architectures for $N_\theta$. With the increasing number of layers, the computation becomes increasingly expensive as the resulting tensor becomes high-dimensional. In order to overcome this, we consider two intermediate or mid-level features. Our choice is guided by the understanding that the initial layers of such deep networks capture generic image features, while the latter layers are often biased towards the pre-training classification task (Roth et al., 2022). We denote the features extracted at a layer $j$ for an image $\mathbf{X}$ as $N_\theta^j(\mathbf{X})$. Following this convention, we express the feature map $\mathbf{U} \in \mathcal{U} = \mathbb{R}^{c^* \times h^* \times w^*}$ produced by the *Feature Aggregator* (FA) as a concatenation of $N_\theta^j(\mathbf{X})$ and $N_\theta^{j+1}(\mathbf{X})$ obtained from layers $j$ and $j+1$ of $N_\theta$. Furthermore, to facilitate the concatenation of features extracted from multiple layers of the extractor $N_\theta$, the features at the lower resolution layer $j+1$ are linearly rescaled by FA to match the dimension of the features at layer $j$. We define an invertible transformation $f : \mathbb{R}^{c^* \times h^* \times w^*} \to \mathbb{R}^{c^* \times k^*}$ where $k^* = h^* \times w^*$ to convert tensor to matrix and vice versa using $f^{-1}$. The function $f$ can be computed in practice by reshaping the tensor to obtain a 2D matrix. Now, using $f$, we compute $\mathbf{Z} = f(\mathbf{U})$. We denote each patch feature within the feature map $\mathbf{Z}$ by $\mathbf{z}_k = \mathbf{Z}[:, k] = \mathbf{U}[:, h, w]$, where $k = (h-1) \times w^* + w$, $h \in \{1, 2, \ldots, h^*\}$, $w \in \{1, 2, \ldots, w^*\}$.

### 3.2 Detecting Structural Anomalies

Having defined $\mathbf{Z}$ in the previous section, we elaborate on the local branch of `ULSAD` for the detection of subtle localized defects in the images, i.e. structural anomalies. Specifically, our goal is to learn the reconstruction of the patch features using the dataset $\mathcal{D}_N$ composed of only normal images. Therefore, we can identify the structural anomalies when the network fails to reconstruct a patch feature during inference.

**Feature Reconstruction Network (FRN)**. As shown in Figure 3, `ULSAD` utilizes a convolutional encoder-decoder architecture with a lower-dimensional bottleneck for learning to reconstruct the feature map $\mathbf{U}$ using the training dataset $\mathcal{D}_N$. First, the encoder network $N_{\psi_e}$ compresses the feature $\mathbf{U}$ to a lower dimensional space, which induces the information bottleneck. It acts as an implicit regularizer, preventing generalization to features corresponding to anomalous images. The encoded represen-

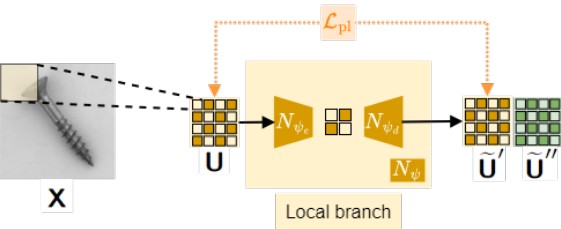

Figure 3: Feature Reconstruction Network

tation is then mapped back to the latent space using a decoder network $N_{\psi_d}$. The output of FRN is $\widetilde{\mathbf{U}} = N_\psi(\mathbf{U}) \in \mathbb{R}^{2c^* \times h^* \times w^*}$ where $N_\psi = N_{\psi_e} \circ N_{\psi_d}$. Besides using the FRN to learn the reconstruction of the patch features for the detection of structural anomalies, we also utilize it to reduce errors during the detection of logical anomalies, as discussed in Section 3.3. To minimizing computation costs and avoid the use of two separate FRNs, we adopt a shared FRN. This is achieved by doubling the number of output channels in the decoder to simultaneously produce two feature maps $\widetilde{\mathbf{U}}'$ and $\widetilde{\mathbf{U}}''$ for the detection of structural and logical anomalies, respectively, with both having dimension $c^* \times h^* \times w^*$.

Although the feature maps $\mathbf{U}$ have significantly lower dimensionality compared to the input images $\mathbf{X}$, they can still be considered high-dimensional tensors. In high-dimensional spaces, the $\ell_2$ distance is not effective at distinguishing between the nearest and furthest points (Aggarwal et al., 2001), making it an inadequate measure for computing the difference between feature maps during training. Therefore, similar to Salehi et al. (2021), we propose combining $\ell_2$ and cosine distances to account for differences in both the magnitude and direction of the patch features as:

$$\mathcal{L}_{\text{pl}}(\widetilde{\mathbf{Z}}', \mathbf{Z}) = \frac{1}{k^*} \sum_{k=1}^{k^*} l_v(\widetilde{\mathbf{z}}'_k, \mathbf{z}_k) + \lambda_l \, l_d(\widetilde{\mathbf{z}}'_k, \mathbf{z}_k), \tag{1}$$

where $\widetilde{\mathbf{Z}}' = f(\widetilde{\mathbf{U}}')$, $\mathbf{Z} = f(\mathbf{Z})$ and $\lambda_l \geq 0$ controls the effect of $l_d$. Furthermore, $l_v(\widetilde{\mathbf{z}}'_k, \mathbf{z}_k)$ and $l_d(\widetilde{\mathbf{z}}'_k, \mathbf{z}_k)$ measure the differences in magnitude and direction between the patch features $\mathbf{z}_k$ and $\widetilde{\mathbf{z}}'_k$, respectively, as

$$l_v(\widetilde{\mathbf{z}}'_k, \mathbf{z}_k) = \|\widetilde{\mathbf{z}}'_k - \mathbf{z}_k\|_2^2, \quad \text{and} \quad l_d(\widetilde{\mathbf{z}}'_k, \mathbf{z}_k) = 1 - \frac{(\widetilde{\mathbf{z}}'_k)^T \mathbf{z}_k}{\|\widetilde{\mathbf{z}}'_k\|_2 \|\mathbf{z}_k\|_2}. \tag{2}$$

### 3.3 Detecting Logical Anomalies

Although the feature reconstruction task discussed in Section 3.2 allows us to detect structural anomalies, it is not suited for identifying logical anomalies that violate the logical constraints of normal images. Recall that such violations appear in the form of misplaced, misaligned, or surplus objects found in normal images. If we consider the example of misaligned objects, the previously discussed approach will fail as it focuses on the individual image patches, which would be normal. It is the overall spatial arrangement of objects in the image which is anomalous. Thus, to identify such anomalies, our goal is to learn the spatial relationships among the objects present in the normal images of the training dataset $\mathcal{D}_N$. We achieve this with the global branch of `ULSAD`, shown in Figure 4, which leverages the entire image and not just its individual patches.

In order to achieve our goal, we start by analyzing the feature maps extracted using the pretrained network $N_\theta$. Pre-trained CNNs tend to have similar activation patterns for semantically similar objects (Tung & Mori, 2019; Zagoruyko & Komodakis, 2017). In Figure 5, we visualize four self-attention maps computed from the features of a pre-trained Wide-Resnet50-2 network. It can be seen that in the first map, all the items for the semantic class "fruits" receive a high attention score. The remaining attention maps focus on individual semantic concepts like "oranges", "cereal" and "plate", respectively. Based on this observation and inspired by the attention-transfer concept for knowledge distillation (Zagoruyko & Komodakis, 2017; Tung & Mori, 2019), we propose to learn the spatial relationships (Dosovitskiy et al., 2021) among the patch features in $\mathbf{U}$ obtained from normal images. Recall that each patch feature corresponds to a patch in the image. Therefore, learning the spatial relationships among the patch features would allow us to learn the spatial relationships among the patches in the image. This forces `ULSAD` to learn the relative positions of objects in

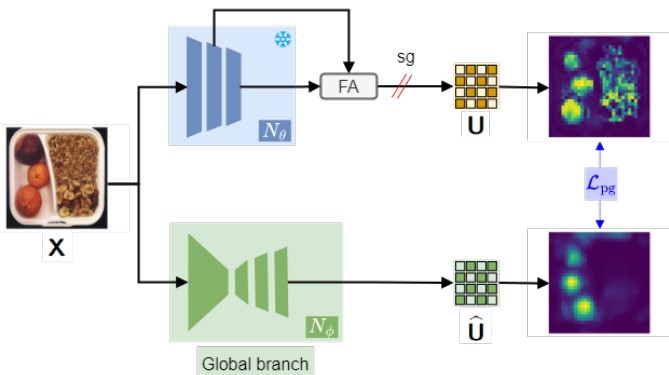

Figure 4: Global Branch of `ULSAD`

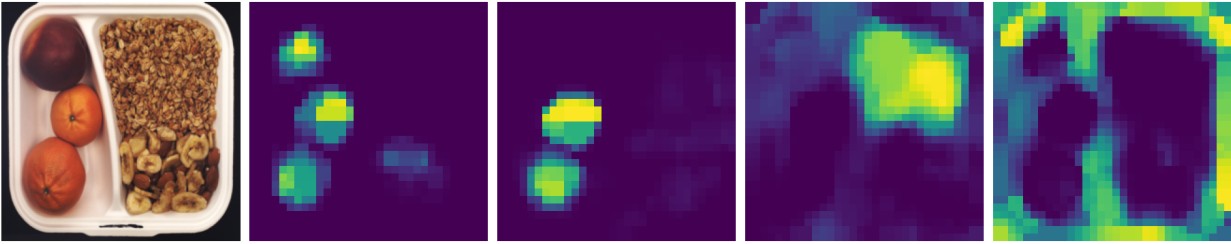

Figure 5: (First) Example image belonging to the category "breakfast box" in the MVTecLOCO dataset. (Rest) Visualization of attention maps computed using the intermediate features from a pre-trained model.

the normal images, thereby enabling it to capture the logical constraints. Starting from $\boldsymbol{Z} = f(\mathbf{U})$, we first compute the self-attention weight matrix $\boldsymbol{W} \in \mathbb{R}^{k^* \times k^*}$ as:

$$\boldsymbol{W}[p, q] = \frac{\exp(\boldsymbol{z}_p^T \boldsymbol{z}_q / \sqrt{c^*})}{\sum_{k=1}^{k^*} \exp(\boldsymbol{z}_k^T \boldsymbol{z}_q / \sqrt{c^*})}. \tag{3}$$

Then, the attention map $\boldsymbol{A} \in \mathbb{R}^{c^* \times k^*}$ is computed as $\boldsymbol{A} = \boldsymbol{ZW}$. For learning the spatial relations using $\boldsymbol{A}$ as our target, we use a convolutional autoencoder-like network $N_\phi = N_{\phi_e} \circ N_{\phi_d}$ where $N_{\phi_e}$ is the encoder and $N_{\phi_d}$ is the decoder. Similar to a standard autoencoder, $N_{\phi_e}$ compresses the input image $\mathbf{X}$ to a lower dimensional space. However, $N_{\phi_d}$ maps the encoded representation to the feature space $\mathcal{U}$, which has a lower dimension than the input space $\mathcal{X}$. We denote the output of $N_\phi$ as $\widehat{\mathbf{U}} = N_\phi(\mathbf{X})$.

A direct approach would be to compute the self-attention map for $\widehat{\mathbf{U}}$ and minimize its distance from $\boldsymbol{A}$. However, it makes the optimization problem computationally challenging as each vector in $\widehat{\mathbf{U}}$ is coupled with every other vector by the network weights $N_\phi$ (Zhang et al., 2023). To overcome this, we compute the cross-attention map $\widehat{\boldsymbol{A}} \in \mathbb{R}^{c^* \times k^*}$ between $\mathbf{U}$ and $\widehat{\mathbf{U}}$. Given $\widehat{\mathbf{Z}} = f(\widehat{\mathbf{U}})$, we first compute $\widehat{\boldsymbol{W}}$ as:

$$\widehat{\boldsymbol{W}}[p, q] = \frac{\exp(\boldsymbol{z}_p^T \widehat{\boldsymbol{z}}_q / \sqrt{c^*})}{\sum_{k=1}^{k^*} \exp(\boldsymbol{z}_k^T \widehat{\boldsymbol{z}}_q / \sqrt{c^*})}. \tag{4}$$

Then, the attention map $\widehat{\boldsymbol{A}}$ can be computed as $\widehat{\boldsymbol{A}} = \boldsymbol{Z}\widehat{\boldsymbol{W}}$. Given, the self-attention map $\boldsymbol{A}$ and the cross-attention map $\widehat{\boldsymbol{A}}$, we define a consistency loss $\mathcal{L}_{\mathrm{pg}}$ as:

$$\mathcal{L}_{\mathrm{pg}}(\widehat{\boldsymbol{A}}, \boldsymbol{A}) = \frac{1}{k^*} \sum_{k=1}^{k*} l_v(\widehat{\boldsymbol{a}}_k, \boldsymbol{a}_k) + \lambda_g \, l_d(\widehat{\boldsymbol{a}}_k, \boldsymbol{a}_k), \tag{5}$$

where $\boldsymbol{a}_k = \boldsymbol{A}[:, k]$, $\widehat{\boldsymbol{a}}_k = \widehat{\boldsymbol{A}}[:, k]$ and $\lambda_{\mathrm{g}} \geq 0$ controls the effect of $l_d$. A limitation of this approach is that autoencoders usually struggle with generating fine-grained patterns as also observed by prior works (Dosovitskiy & Brox, 2016; Assran et al., 2023). As a result, the global branch is prone to false positives in the presence of sharp edges or heavily textured surfaces due to the loss of high-frequency details. To address this limitation, we utilize the FRN $N_\psi$ in the local branch to learn the output $\widehat{\mathbf{U}}$. Recall that the output of FRN $\widetilde{\mathbf{U}} \in \mathbb{R}^{2c^* \times h^* \times w^*}$ has $2c^*$ number of channels to simultaneously generate two feature maps $\widetilde{\mathbf{U}}'$ and $\widetilde{\mathbf{U}}''$, both having dimension $c^* \times h^* \times w^*$. Out of which, $\widetilde{\mathbf{U}}'$ is used for learning the patch features. Here, we define the loss $\mathcal{L}_{lg}$ to relate the local feature map $\widetilde{\mathbf{U}}''$ with the global feature map $\widehat{\mathbf{U}}$ as:

$$\mathcal{L}_{\mathrm{lg}}(\widetilde{\boldsymbol{Z}}'', \widehat{\boldsymbol{Z}}) = \frac{1}{k^*} \sum_{k=1}^{k*} l_v(\widetilde{\boldsymbol{z}}_k'', \widehat{\boldsymbol{z}}_k) + \lambda_g \, l_d(\widetilde{\boldsymbol{z}}_k'', \widehat{\boldsymbol{z}}_k), \tag{6}$$

where $\widetilde{\boldsymbol{Z}}'' = f(\widetilde{\mathbf{U}}'')$. Therefore, during inference, a difference between the $\widetilde{\mathbf{U}}''$ and $\widehat{\mathbf{U}}$ indicates the presence of logical anomalies. The benefits of such a framework are two-fold: (1) it allows for learning the spatial relationships in the normal images while reducing the chance of having false positives, and (2) doubling the channels in the decoder allows sharing the encoder architecture, reducing the computational costs.

---

**Algorithm 1:** Unified Logical and Structural AD (`ULSAD`) //  Local branch   Global branch

---

**Require:** Training dataset $\mathcal{D}_N$, Feature extractor $N_\theta$, Feature reconstruction network $N_\psi$

Global autoencoder $N_\phi$, Number of epochs $e$, Learning rate $\eta$, Pre-trained feature statictics $\boldsymbol{\mu}$, $\boldsymbol{\sigma}$

**1** **for** (epoch $\in 1, \cdots, e$) and ($\mathbf{X} \in \mathcal{D}_N$) **do**

**2**  Extract normalized features maps using the pre-trained network:

**3**   $\mathbf{U} \leftarrow N_\theta(\mathbf{X})$

**4**   $\mathbf{U} \leftarrow (\mathbf{U} - \boldsymbol{\mu})/\boldsymbol{\sigma}$

**5**   $\boldsymbol{Z} \leftarrow f(\mathbf{U})$

**6**  Reconstruct the features maps using the local branch:

**7**   $\widetilde{\mathbf{U}} \leftarrow N_\psi(\mathbf{U})$

**8**   $\widetilde{\boldsymbol{Z}} \leftarrow f(\widetilde{\mathbf{U}})$

**9**  Compute local loss (Eq. 1):

**10**   $l_l \leftarrow \mathcal{L}_{\mathrm{pl}}(\widetilde{\boldsymbol{Z}}', \boldsymbol{Z})$

**11**  Obtain the output of the global autoencoder:

**12**   $\widehat{\mathbf{U}} \leftarrow N_\phi(\mathbf{X})$

**13**   $\widehat{\boldsymbol{Z}} \leftarrow f(\widehat{\mathbf{U}})$

**14**  Compute consistency loss (Eq. 5):

**15**   $l_g \leftarrow \mathcal{L}_{pg}(\widehat{\boldsymbol{Z}}, \boldsymbol{Z})$

**16**  Compute local-global loss (Eq. 6):

**17**   $l_{\mathrm{lg}} \leftarrow \mathcal{L}_{lg}(\widehat{\boldsymbol{Z}}, \widetilde{\boldsymbol{Z}}'')$

**18**  Compute overall loss:

**19**   $l \leftarrow l_l + l_g + l_{\mathrm{lg}}$

**20**  Update model parameters:

**21**   $\psi \leftarrow \psi - \eta\nabla_\psi l$

**22**   $\phi \leftarrow \phi - \eta\nabla_\phi l$

**23** **end**

**Return:** $N_\psi$, $N_\phi$

---

### 3.4 `ULSAD` Algorithm Overview

An overview of `ULSAD` is outlined in Algorithm 1, which can simultaneously detect structural and logical anomalies. Firstly, we pass a normal image $\mathbf{X}$ from the training dataset $\mathcal{D}_N$ through the feature extractor $N_\theta$ to obtain feature maps $\mathbf{U}$. We normalize the features (line 4, Algorithm 1) with the channel-wise mean $\boldsymbol{\mu}$ and standard deviation $\boldsymbol{\sigma}$ computed over all the images in $\mathcal{D}_N$. We do not include this step in Algorithm 1 as the calculation is trivial. Instead, we consider the values $\boldsymbol{\mu}$ and $\boldsymbol{\sigma}$ to be given as input parameters for the sake of simplicity. Secondly, we obtain $\widetilde{\mathbf{U}}$ by passing $\mathbf{U}$ through the feature reconstruction network $N_\psi$ (line 7, Algorithm 1). Recall that, $\widetilde{\mathbf{U}}$ has a dimension $2c^* \times h^* \times w^*$ which can be decomposed into two feature maps $\widetilde{\mathbf{U}}'$ and $\widetilde{\mathbf{U}}''$ each with a dimension $c^* \times h^* \times w^*$. The feature reconstruction loss $\mathcal{L}_{\mathrm{pl}}$ is then computed between $\boldsymbol{Z}$ and $\widetilde{\boldsymbol{Z}}'$, where $\boldsymbol{Z} = f(\mathbf{U})$ and $\widetilde{\boldsymbol{Z}}' = f(\widetilde{\mathbf{U}}')$. Thirdly, we obtain the features $\widehat{\mathbf{U}}$ by passing the input sample $\mathbf{X}$ through the autoencoder $N_\phi$. Then for learning the spatial relationships from the normal images, we compute $\mathcal{L}_{\mathrm{pg}}$ between the self-attention map of $\boldsymbol{Z}$ and the cross-attention map between $\boldsymbol{Z}$ and $\widehat{\boldsymbol{Z}} = f(\widehat{\mathbf{U}})$ (line 15, Algorithm 1). In the fourth step, we compute the loss $\mathcal{L}_{\mathrm{lg}}$ between $\widehat{Z}$ and $\widetilde{\boldsymbol{Z}}'' = f(\widetilde{\mathbf{U}}'')$. Finally, the model parameters $\psi$ and $\phi$ are updated based on the gradient of the total loss (line $21-22$, Algorithm 1). The end-to-end pipeline is illustrated in Figure 2.

### 3.5 Anomaly Detection and Localization

After discussing how `ULSAD` is trained to detect structural and logical anomalies, we now focus on the inference process. The first step is to compute an anomaly map $\boldsymbol{M}$ for a given test image $\mathbf{X}$, which assigns a per-pixel

anomaly score. We begin by calculating the local anomaly map $\boldsymbol{M}^l \in \mathbb{R}^{h^* \times w^*}$ based on the difference between the output of the local branch $\widetilde{\mathbf{U}}'$ and the feature map $\mathbf{U}$, as follows:

$$\boldsymbol{M}^l[h, w] = l_v(\widetilde{\mathbf{U}}'[:, h, w], \ \mathbf{U}[:, h, w]) + \lambda_l \ l_d(\widetilde{\mathbf{U}}'[:, h, w], \ \mathbf{U}[:, h, w]), \tag{7}$$

where $\widetilde{\mathbf{U}}' = f^{-1}(\widetilde{\boldsymbol{Z}}')$ and $\mathbf{U} = f^{-1}(\boldsymbol{Z})$. Similarly, the global anomaly map $\boldsymbol{M}^g$ is computed using the output from the global autoencoder $\widehat{\mathbf{U}}$ and the local reconstruction branch $\widetilde{\mathbf{U}}''$:

$$\boldsymbol{M}^g[h, w] = l_v(\widetilde{\mathbf{U}}''[:, h, w], \widehat{\mathbf{U}}[:, h, w]) + \lambda_g \ l_d(\widetilde{\mathbf{U}}''[:, h, w], \widehat{\mathbf{U}}[:, h, w]), \tag{8}$$

where $\widetilde{\mathbf{U}}'' = f^{-1}(\widetilde{\boldsymbol{Z}}'')$ and $\widehat{\mathbf{U}} = f^{-1}(\widehat{\boldsymbol{Z}})$.

Since $\boldsymbol{M}^l$ and $\boldsymbol{M}^g$ may have different ranges of anomaly scores, we normalize each map independently. This normalization ensures consistent score ranges and prevents noise in one map from overwhelming anomalies detected in the other. Given the variability in anomaly score distributions across datasets, we adopt a quantile-based normalization method, which makes no assumptions about the underlying score distribution.

To normalize the maps, we generate two sets of anomaly maps: $\mathcal{M}^l = \{\boldsymbol{M}^l \mid \mathbf{X} \in \mathcal{D}_{\text{valid}}\}$ and $\mathcal{M}^g = \{\boldsymbol{M}^g \mid \mathbf{X} \in \mathcal{D}_{\text{valid}}\}$, using images from the validation set $\mathcal{D}_{\text{valid}}$. For each set, we pool together the pixel values from all the anomaly maps in that set to compute the empirical quantiles at significance levels $\alpha$ and $\beta$. Specifically, for the local anomaly maps, the quantiles are denoted as $q_\alpha^l$ and $q_\beta^l$, while for the global anomaly maps, the quantiles are denoted as $q_\alpha^g$ and $q_\beta^g$. Values below $q_\alpha$ are considered normal, while those above $q_\beta$ are marked as highly abnormal.

Following Batzner et al. (2024), we define linear transformations $t^l(\cdot)$ and $t^g(\cdot)$ for the local and global anomaly maps to map normal pixels to values $\leq 0$ and highly anomalous pixels to values $\geq 0.1$:

$$t^l(\boldsymbol{M}^l) = 0.1 \left( \boldsymbol{M}^l - \left( \frac{q_\alpha^l}{q_\beta^l - q_\alpha^l} \right) \mathbf{1}_{h^* \times w^*} \right), \quad t^g(\boldsymbol{M}^g) = 0.1 \left( \boldsymbol{M}^g - \left( \frac{q_\alpha^g}{q_\beta^g - q_\alpha^g} \right) \mathbf{1}_{h^* \times w^*} \right),$$

where $\mathbf{1}_{h^* \times w^*}$ is a matrix of ones. Mapping the empirical quantiles at $\alpha$ and $\beta$ to values of 0 and 0.1 helps highlight the anomalous regions on a 0-to-1 color scale for visualization. Normal pixels are assigned a score of 0, while pixels with scores between $q_\alpha$ and $q_\beta$ gradually increase in color intensity. Pixels with scores exceeding $q_\beta$ change more rapidly toward 1. Note that this transformation does not affect AU-ROC scores, as these depend only on the ranking of the scores.

Finally, we compute the overall anomaly map $\boldsymbol{M}$ for the image $\mathbf{X}$ by averaging the normalized local and global maps:

$$\boldsymbol{M} = \frac{t^l(\boldsymbol{M}^l) + t^g(\boldsymbol{M}^g)}{2}.$$

The final anomaly score for $\mathbf{X}$ is the maximum value in the combined anomaly map:

$$s = \max_{h \in \{1, 2, \ldots, h^*\}, w \in \{1, 2, \ldots, w^*\}} \boldsymbol{M}(h, w).$$

## 4 Experimental Evaluation

In this section, we answer the following three questions: (i) How does ULSAD perform as compared to the SOTA methods? (ii) How effective is the local and global branch for the detection of *structural* and *logical* anomalies? (iii) How does each component in ULSAD impact the overall performance?

### 4.1 Setup

**Benchmark Datasets**. We evaluate our method on the following five IAD benchmarking datasets:

**[1] BTAD** (Mishra et al., 2021). It comprises real-world images of three industrial products, with anomalies such as body and surface defects. Training data includes 1,799 normal images across the three categories,

while the test set contains 290 anomalous and 451 normal images.

[**2**] **MVTec AD** (Bergmann et al., 2019). It consists of images from industrial manufacturing across 15 categories comprised of 10 objects and 5 textures. In totality, it contains 3,629 normal images for training. For evaluation, 1,258 anomalous images with varying pixel level defects and 467 normal images.

[**3**] **MVTec-Loco** (Bergmann et al., 2022). An extension of MVTec dataset, it encompasses both local structural anomalies and logical anomalies violating long-range dependencies. It consists of 5 categories, with 1,772 normal images for training and 304 normal images for validation. It also contains 1568 images, either normal or anomalous, for evaluation.

[**4**] **MPDD** (Jezek et al., 2021). It focuses on metal part fabrication defects. The images are captured in variable spatial orientation, position, and distance of multiple objects concerning the camera at different light intensities and with a non-homogeneous background. It consists of 6 classes of metal parts with 888 training images. For evaluation, the dataset has 176 normal and 282 anomalous images.

[**5**] **VisA** (Zou et al., 2022). It contains 10,821 high-resolution images (9,621 normal and 1,200 anomalous images) across 12 different categories. The anomalous images contain different types of anomalies such as scratches, bent, cracks, missing parts or misplacements. For each type of defect, there are 15-20 images, and an image can depict multiple defects.

**Evaluation metrics**. We measure the image-level anomaly detection performance via the area under the receiver operator curve (AUROC) based on the assigned anomaly score. To measure the anomaly localization performance, we use pixel-level AUROC and area under per region overlap curve (AUPRO). Furthermore, following prior works (Roth et al., 2022; Gudovskiy et al., 2021; Bergmann et al., 2019), we compute the average metrics over all the categories for each of the benchmark datasets. Moreover, for ULSAD, we report all the results over 5 runs with different random seeds.

**Baselines**. We compare our method with existing state-of-the-art unsupervised AD methods, namely PatchCore (Roth et al., 2022), PaDim (Defard et al., 2021), CFLOW (Gudovskiy et al., 2021), FastFLOW (Yu et al., 2021), DRÆM (Zavrtanik et al., 2021), Reverse Distillation (RD) (Deng & Li, 2022), EfficientAD (Batzner et al., 2024) and DFR (Yang et al., 2020). In this study, we only consider baselines that are capable of both anomaly detection and localization.

**Implementation details**. ULSAD is implemented in PyTorch (Paszke et al., 2019). For the baselines, we follow the implementation in Anomalib (Akcay et al., 2022), a widely used AD library for benchmarking. In ULSAD, we use a Wide-ResNet50-2 pre-trained on ImageNet (Zagoruyko & Komodakis, 2016) and extract features from the second and third layers, similar to PathCore (Roth et al., 2022). We use a CNN for the autoencoder $N_\phi$ in the global branch and the feature reconstruction network $N_\psi$ in the local branch. It consists of convolution layers with LeakyReLU activation in the encoder and deconvolution layers in the decoder. The architecture is provided in the Appendix A. Unless otherwise stated, for all the experiments, we consider an image size of $256 \times 256$. We train ULSAD over 200 epochs for each category using an Adam optimizer with a learning rate of 0.0002 and a weight decay of 0.00002. We set $\alpha = 0.9$ and $\beta = 0.995$ unless specified otherwise. For the baselines, we use the hyperparameters mentioned in the respective papers.

## 4.2 Evaluation Results

We summarize the anomaly detection performance of ULSAD in Table 1 and the localization performance in Table 2. On the BTAD dataset, we improve over the DFR by approximately 2% in detection. Inspecting the images from the dataset, we hypothesize that the difference stems from the use of a global branch in ULSAD as the structural imperfections are not limited to small regions. For localization, Reverse Distillation performs better owing to the use of anomaly maps computed per layer of the network. We can observe similar improvements over DFR on the MVTec dataset. Although PatchCore provides superior performance on MVTec, it should be noted that even without using a memory bank, ULSAD provides comparable results. Then, we focus on more challenging datasets such as MPDD and MVTecLOCO. While MPDD contains varying external conditions such as lighting, background and camera angles, MVTecLOCO contains both logical and structural anomalies. We can observe improvements over DFR ($\sim 12 - 16\%$) in both datasets. This

highlights the effectiveness of our method. We visualize the anomaly maps for samples from the "pushpin" and "juice bottle" categories in Figure 6. It can be seen that while the global branch is more suited to the detection of logical anomalies, the local branch is capable of detecting localized structural anomalies.

Table 1: Average Detection Performance in AUROC (%). Style: **best** and second best

| Method | BTAD | MPDD | MVTec | MVTec-LOCO | VisA |
|---|---|---|---|---|---|
| PatchCore (Roth et al., 2022) | 93.27 | 93.27 | **98.75** | 81.49 | 91.48 |
| CFLOW (Gudovskiy et al., 2021) | 93.57 | 87.11 | 94.47 | 73.62 | 87.77 |
| DRÆM (Zavrtanik et al., 2021) | 73.42 | 74.14 | 75.26 | 62.35 | 77.75 |
| EfficientAD (Batzner et al., 2024) | 88.26 | 85.42 | 98.23 | 80.62 | 91.21 |
| FastFlow (Yu et al., 2021) | 91.68 | 65.03 | 90.72 | 71.00 | 87.49 |
| PaDiM (Defard et al., 2021) | 93.20 | 68.48 | 91.25 | 68.38 | 83.28 |
| Reverse Distillation (Deng & Li, 2022) | 83.87 | 79.62 | 79.65 | 61.56 | 86.24 |
| DFR (Yang et al., 2020) | 94.60 | 79.75 | 93.54 | 72.87 | 85.18 |
| ULSAD (Ours) | **96.17** ± 0.45 | **95.73** ± 0.45 | 97.65 ± 0.38 | **84.1** ± 0.86 | **92.46** ± 0.45 |

Table 2: Average Segmentation Performance in AUROC (%) and AUPRO (%). Style: **best** and second best

| Method | BTAD | MPDD | MVTec | MVTec-LOCO | VisA |
|---|---|---|---|---|---|
| PatchCore (Roth et al., 2022) | 96.85 \| 71.48 | **98.07** \| 90.84 | **97.71** \| 91.15 | 75.77 \| 69.09 | 97.93 \| 85.12 |
| CFLOW (Gudovskiy et al., 2021) | 96.60 \| 73.11 | 97.42 \| 88.56 | 97.17 \| 90.14 | 76.99 \| 66.93 | 98.04 \| 85.29 |
| DRÆM (Zavrtanik et al., 2021) | 59.04 \| 22.48 | 86.96 \| 70.04 | 75.01 \| 49.72 | 63.69 \| 40.06 | 71.31 \| 54.68 |
| EfficientAD (Batzner et al., 2024) | 82.13 \| 54.37 | 97.03 \| 90.44 | 96.29 \| 90.11 | 70.36 \| 66.96 | 97.51 \| 84.45 |
| FastFlow (Yu et al., 2021) | 96.15 \| 75.27 | 93.60 \| 76.89 | 96.44 \| 88.79 | 75.55 \| 53.04 | 97.32 \| 81.70 |
| PaDiM (Defard et al., 2021) | 97.07 \| 77.80 | 94.51 \| 81.18 | 96.79 \| 91.17 | 71.32 \| 67.97 | 97.09 \| 80.80 |
| Reverse Distillation (Deng & Li, 2022) | **97.85** \| **81.47** | 97.83 \| 91.86 | 97.25 \| **93.12** | 68.55 \| 66.28 | **98.68** \| **91.77** |
| DFR (Yang et al., 2020) | 97.62 \| 59.06 | 97.33 \| 90.46 | 94.93 \| 89.42 | 61.72 \| 69.78 | 97.90 \| 91.72 |
| ULSAD (Ours) | 96.73 \| 75.41 ± 0.51 \| ± 3.95 | 97.45 \| **92.02** ± 0.99 \| ± 2.64 | 97.61 \| 91.67 ± 0.64 \| ± 1.36 | **80.06** \| **73.73** ± 0.20 \| ± 0.35 | 98.24 \| 87.12 ± 0.20 \| ± 0.89 |

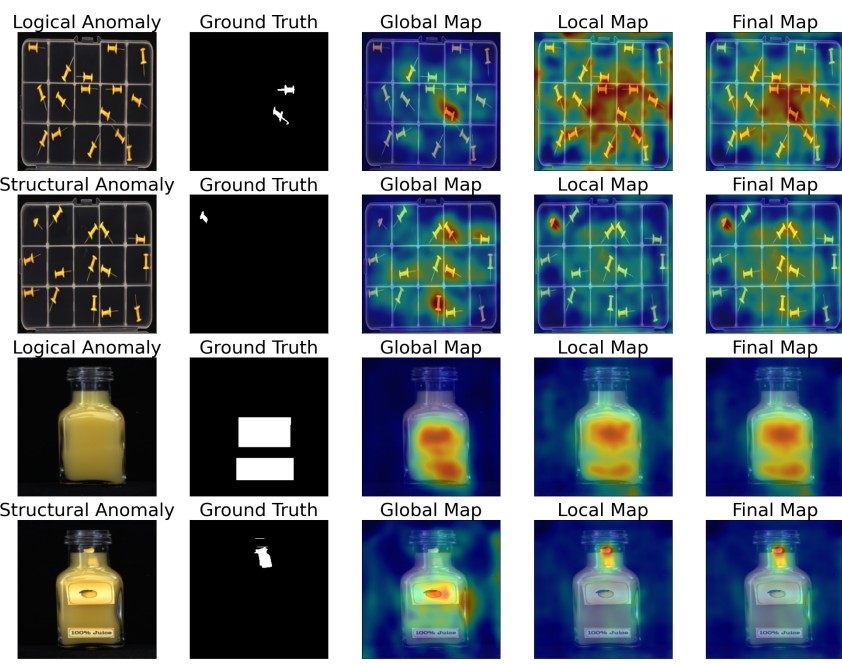

Figure 6: Example of anomaly maps obtained from global and local branches along with the combined map. Overall, ULSAD demonstrates competitive results in anomaly detection compared to the baseline methods across all benchmark datasets. Additionally, the difference in performance between ULSAD and the baselines

for anomaly localization is minimal. The most notable difference is in the AUPRO score on the BTAD, MVTec, and VisA datasets. Nonetheless, while the SOTA methods provide slightly better performance in the localization of structural anomalies, ULSAD provides similar performance across both logical and structural anomalies. We present anomaly maps obtained from different methods in Figure 9 of Appendix B. Extended versions of Tables 1 and 2 are provided in Appendix B. Additionally, we provide the results on MVTecLOCO, split between logical and structural anomalies, in Appendix B.1.

## 4.3 Ablation study

In this section, we analyze the impact of the key components of ULSAD, backbone architectures and the choice of $\alpha$ and $\beta$ for normalization, using the MVTecLOCO dataset.

**Analysis of main components**. We investigate the impact of key components in ULSAD as presented in Table 3. Initially, we set both $\lambda_l$ and $\lambda_g$ to 0, focusing solely on differences in magnitude when computing $\mathcal{L}_{pg}$, $\mathcal{L}_{lg}$, and $\mathcal{L}_{pl}$. The first row corresponds to using only the local branch. In the third row, the consistency loss $\mathcal{L}_{pg}$ is applied to capture spatial relationships for detecting logical anomalies. However,

Table 3: Ablation of the main components of ULSAD.

| | Local Branch | Global Branch | | | | Performance (%) |
|---|---|---|---|---|---|---|
| | $\lambda_l$ | $\lambda_g$ | $\mathcal{L}_{lg}$ | $\mathcal{L}_{pg}^d$ | $\mathcal{L}_{pg}$ | I-AUROC \| P-AUROC \| P-AUPRO |
| 1 | 0.0 | - | - | - | - | 77.67 \| 75.17 \| 73.37 |
| 2 | 0.0 | 0.0 | - | ✓ | - | 77.69 \| 79.77 \| 75.26 |
| 3 | 0.0 | 0.0 | - | - | ✓ | 71.67 \| 73.92 \| 67.22 |
| 4 | 0.0 | 0.0 | ✓ | ✓ | - | 81.40 \| 82.12 \| 77.47 |
| 5 | 0.0 | 0.0 | ✓ | - | ✓ | 81.08 \| 81.97 \| 76.45 |
| 6 | 0.5 | - | - | - | - | 79.14 \| 76.57 \| 73.41 |
| 7 | 0.5 | 0.5 | - | ✓ | - | 80.50 \| 81.85 \| 77.35 |
| 8 | 0.5 | 0.5 | - | - | ✓ | 74.51 \| 76.59 \| 69.01 |
| 9 | 0.5 | 0.5 | ✓ | ✓ | - | 82.19 \| 81.25 \| 75.50 |
| 10 | 0.5 | 0.5 | ✓ | - | ✓ | **84.10** \| **80.06** \| **73.73** $\pm$ 0.86 \| $\pm$ 0.20 \| $\pm$ 0.35 |

when used in isolation, it limits ULSAD's performance to detecting only logical anomalies and fails to capture localized structural anomalies. Additionally, as discussed in Section 3.3, the global branch is prone to false positives in the presence of sharp edges or heavily textured surfaces. When incorporating $\mathcal{L}_{lg}$, which connects the global and local branches, we observe a significant improvement in performance, as shown in the fifth row. For the sake of completeness, we also consider here a variant of the consistency loss $\mathcal{L}_{pg}$ where we compute the $\ell_2$ distance between the feature maps instead of computing the self- and cross-attention maps. We refer to the alternative in the table as $\mathcal{L}_{pg}^d$. We observe that the difference between the two variants becomes negligible when combined with $\mathcal{L}_{lg}$ (row 4 and 5). Further, incorporating differences in direction when computing $\mathcal{L}_{pg}$, $\mathcal{L}_{lg}$, and $\mathcal{L}_{pl}$ leads to improved performances across all settings as shown in the last five rows. Overall, the best performance is obtained when both $\mathcal{L}_{lg}$ and $\mathcal{L}_{pg}$ is used while considering differences in both direction and magnitude for computing the losses.

**Effect of backbone**. We investigate the impact of using different pre-trained backbones in ULSAD in Figure 7. We can observe that the overall best performance is obtained by using a Wide-ResNet101-2 architecture in both detection and localization. More specifically, for detection, Wide-ResNet variants are more effective than the ResNet architectures, whereas, for localization performance measured using Pixel AUROC, the deeper networks such as ResNet152 and Wide-ResNet101-2 seem to have precedence over their shallower counterparts. Overall, we can see that performance is robust to the

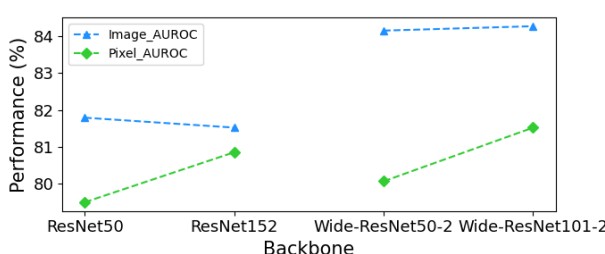

Figure 7: Ablation study of the backbone network

choice of pre-trained model architecture. In our experiments, we utilize a Wide-ResNet50-2 architecture which is used by most of our baselines for fair comparison.

**Effect of normalization**. We analyze the impact of the quantile-based normalization on the performance metrics by considering multiple values for $\alpha$ and $\beta$. The results are shown in Figure 8. It can be seen that the final performance is robust to the choice of $\alpha$ and $\beta$.

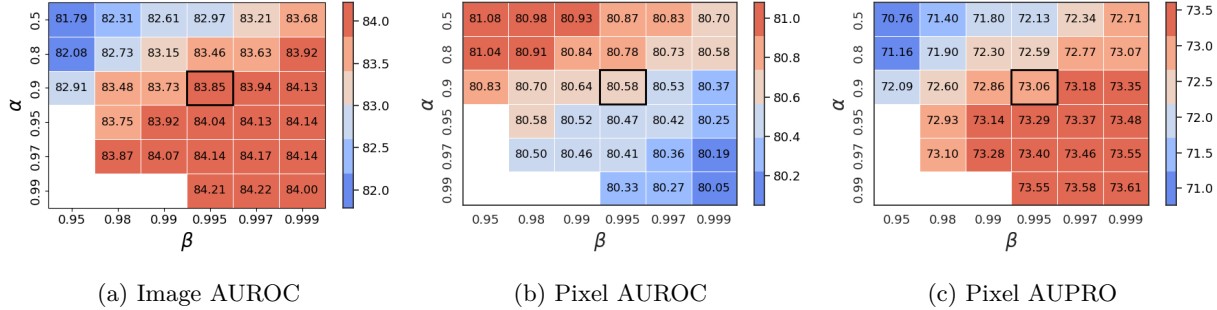

Figure 8: Ablation study of $\alpha$ and $\beta$ for normalization of anomaly maps with selected value highlighted.

Table 4: Memory and Computational Efficiency on MVTecLOCO dataset.

| | CFLOW (2021) | DRÆM (2021) | FastFlow (2021 | PaDiM (2021) | PatchCore (2022) | RD (2022) | DFR (2020) | EffAD (2024) | ULSAD (Ours) |
|---|---|---|---|---|---|---|---|---|---|
| I-AUROC ↑ | 73.62 | 62.35 | 71.00 | 68.38 | 81.49 | 61.56 | 72.87 | 80.62 | **84.10** ± 0.86 |
| P-AUROC ↑ | 76.99 | 63.69 | 75.55 | 71.32 | 75.77 | 68.55 | 61.72 | 70.36 | **80.06** ± 0.20 |
| P-AUPRO ↑ | 66.93 | 40.06 | 53.04 | 67.97 | 69.09 | 66.28 | 69.78 | 66.96 | **73.73** ± 0.35 |
| Throughput (img / s) ↑ | 11.69 | 10.06 | 30.21 | 33.45 | 32.70 | **34.87** | 15.08 | 23.33 | 33.42 |
| GPU Memory (GB) ↓ | 2.57 | 7.95 | **1.69** | 1.92 | 6.80 | 1.93 | 5.85 | 3.48 | 2.17 |

## 5 Memory and Computational Complexity

We report the computational cost and memory requirements of ULSAD compared to the baselines in Table 4. For this analysis, we ran inference on the test samples in the MVTecLOCO dataset using an NVIDIA A100 GPU. We measured throughput with a batch size of 32, as a measure of computational complexity, following EfficientAD (Batzner et al., 2024). Throughput is defined as the number of images processed per second when processing in batches. ULSAD demonstrates higher throughput than most baselines while maintaining competitive anomaly detection and localization performance. In addition to throughput, we also report peak GPU memory usage in Table 4 to highlight the memory efficiency of ULSAD. It is evident that ULSAD requires approximately one-third of the memory compared to retrieval-based methods such as PatchCore, which is one of the state-of-the-art methods for IAD. For DFR (Yang et al., 2020), we follow the authors' approach by using a multiscale representation, concatenating features from 12 layers of the pre-trained network $N_\theta$ for anomaly detection. This approach results in reduced throughput and increased memory usage, as shown in Table 4. With our proposed modifications in ULSAD, we achieve superior performance using features from only 2 layers of $N_\theta$, drastically reducing memory requirements to approximately one-third of DFR's and increasing throughput by approximately two times.

## 6 Limitations

For training ULSAD, we follow the common assumption in unsupervised anomaly detection (Ruff et al., 2021; Chandola et al., 2009; Roth et al., 2022; Batzner et al., 2024) that the training dataset is "clean", meaning it contains no anomalous samples. This setup is known in the literature as one-class classification (Ruff et al., 2018). However, this assumption could impact performance in real-world scenarios where anomalies are unknown a priori. Investigating the effects of dataset contamination (Wang et al., 2019; Jiang et al., 2022; Yoon et al., 2022; Perini et al., 2023; 2022) is an active area of research, which is beyond the scope of our current work. We leave for future research the analysis of contamination's impact on ULSAD and the development of strategies to make the learning process robust in the presence of anomalies.

## 7 Conclusion

Our study focuses on Deep Feature Reconstruction (DFR), a memory- and compute-efficient method for detecting structural anomalies. We propose ULSAD, a unified framework that extends DFR to detect both structural and logical anomalies using a dual-branch architecture. In particular, we enhance the local branch's training objective to account for differences in the magnitude and direction of patch features, thereby improving

structural anomaly detection. Additionally, we introduce an attention-based loss in the global branch to capture logical anomalies effectively. Extensive experiments on five benchmark image anomaly detection datasets demonstrate that `ULSAD` achieves competitive performance in anomaly detection and localization compared to eight state-of-the-art methods. Notably, `ULSAD` also performs well against memory-intensive, retrieval-based methods like PatchCore (Roth et al., 2022). Finally, ablation studies highlight the impact of various components in `ULSAD` and the role of the pre-trained backbone on overall performance.

**Acknowledgment**

This work is supported by the FLARACC research project (Federated Learning and Augmented Reality for Advanced Control Centers), funded by the Wallonia region in Belgium

**Reproducibility Statement**

We provide extensive descriptions of implementation details (in Section 4.1), algorithm (in Algorithm 1) and code to help readers reproduce our results. Every measure is taken to ensure fairness in our comparisons by adopting the most commonly adopted evaluation settings in the anomaly detection literature. More specifically, we use the Anomalib library for our experiments, whenever applicable, for comparing their performance with `ULSAD`. For methods such as DFR (Yang et al., 2020), which is not implemented in Anomalib, we refer to the original codebase provided by the authors.

**Ethics Statement**

We have read the TMLR Ethics Guidelines (https://jmlr.org/tmlr/ethics.html) and ensured that this work adheres to it. All benchmark datasets and pre-trained model checkpoints are publicly available and not directly subject to ethical concerns.

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

# A Implementation Details

ULSAD is implemented in PyTorch (Paszke et al., 2019). Specifically, we used the Anomalib (Akcay et al., 2022) library by incorporating our code within it. It helps us have a fair comparison as we use the implementations of baselines from Anomalib. Moreover, we used a single NVIDIA A4000 GPU for all the experiments unless mentioned otherwise. The architecture of FRN and global autoencoder-like model is provided in Table 5 and 6, respectively.

Table 5: Feature Reconstruction Network of ULSAD.

|  | Layer Name | Stride | Kernel Size | Number of Kernels | Padding | Activation |
|---|---|---|---|---|---|---|
| Encoder | Conv-1 | 2 | $3 \times 3$ | 768 | 1 | ReLU |
|  | BatchNorm-1 | - | - | - | - | - |
|  | Conv-2 | 2 | $3 \times 3$ | 1536 | 1 | ReLU |
|  | BatchNorm-2 | - | - | - | - | - |
|  | Conv-3 | 1 | $3 \times 3$ | 1536 | 1 | ReLU |
|  | BatchNorm-3 | - | - | - | - | - |
| Decoder | ConvTranspose-1 | 2 | $4 \times 4$ | 768 | 1 | ReLU |
|  | BatchNorm-4 | - | - | - | - | - |
|  | ConvTranspose-2 | 2 | $4 \times 4$ | 384 | 1 | ReLU |
|  | BatchNorm-5 | - | - | - | - | - |
|  | ConvTranspose-3 | 1 | $5 \times 5$ | 384 | 1 | ReLU |
|  | BatchNorm-6 | - | - | - | - | - |

Table 6: Global Autoencoder of ULSAD.

|  | Layer Name | Stride | Kernel Size | Number of Kernels | Padding | Activation |
|---|---|---|---|---|---|---|
| Encoder | Conv-1 | 2 | $4 \times 4$ | 32 | 1 | ReLU |
|  | BatchNorm-1 | - | - | - | - | - |
|  | Conv-2 | 2 | $4 \times 4$ | 32 | 1 | ReLU |
|  | BatchNorm-2 | - | - | - | - | - |
|  | Conv-3 | 2 | $4 \times 4$ | 64 | 1 | ReLU |
|  | BatchNorm-3 | - | - | - | - | - |
|  | Conv-4 | 2 | $4 \times 4$ | 64 | 1 | ReLU |
|  | BatchNorm-4 | - | - | - | - | - |
|  | Conv-5 | 2 | $4 \times 4$ | 64 | 1 | ReLU |
|  | BatchNorm-5 | - | - | - | - | - |
|  | Conv-6 | 1 | $8 \times 8$ | 64 | 1 | ReLU |
|  | BatchNorm-6 | - | - | - | - | - |
| Decoder | Interpolate-1 (31, mode= "bilinear") | - | - | - | - | - |
|  | Conv-1 | 1 | $4 \times 4$ | 64 | 2 | ReLU |
|  | BatchNorm-1 | - | - | - | - | - |
|  | Interpolate-2 (8, mode= "bilinear") | - | - | - | - | - |
|  | Conv-2 | 1 | $4 \times 4$ | 64 | 2 | ReLU |
|  | BatchNorm-2 | - | - | - | - | - |
|  | Interpolate-3 (16, mode= "bilinear") | - | - | - | - | - |
|  | Conv-3 | 1 | $4 \times 4$ | 64 | 2 | ReLU |
|  | BatchNorm-3 | - | - | - | - | - |
|  | Interpolate-4 (32, mode= "bilinear") | - | - | - | - | - |
|  | Conv-4 | 1 | $4 \times 4$ | 64 | 2 | ReLU |
|  | BatchNorm-4 | - | - | - | - | - |
|  | Interpolate-5 (64, mode= "bilinear") | - | - | - | - | - |
|  | Conv-5 | 1 | $4 \times 4$ | 64 | 2 | ReLU |
|  | BatchNorm-5 | - | - | - | - | - |
|  | Interpolate-6 (32, mode= "bilinear") | - | - | - | - | - |
|  | Conv-6 | 1 | $3 \times 3$ | 64 | 1 | ReLU |
|  | BatchNorm-6 | - | - | - | - | - |
|  | Conv-7 | 1 | $3 \times 3$ | 384 | 1 | ReLU |

# B Extended Results

Extended versions of the Table 1 and 2 are provided in Tables 7-21. It shows the performance of `ULSAD` per category of the benchmark datasets for anomaly detection and localization. Additionally, we provide a visual comparison of the generated anomaly maps using the MVTecLOCO dataset in Figure 9.

Table 7: Anomaly detection based on Image AUROC on MVTec dataset.

| Category | CFLOW (2021) | DRÆM (2021) | FastFlow (2021 | PaDiM (2021) | PatchCore (2022) | RD (2022) | DFR (2020) | EffAD (2024) | ULSAD (Ours) |
|---|---|---|---|---|---|---|---|---|---|
| bottle | **100.0** | 94.6 | 98.57 | 99.37 | **100.0** | 98.10 | 94.92 | **100.0** | **100.0** ± 0.00 |
| cable | 93.46 | 74.29 | 89.30 | 86.94 | **98.54** | 95.41 | 79.54 | 94.61 | 97.92 ± 0.18 |
| capsule | 91.74 | 65.46 | 86.04 | 88.43 | **97.93** | 81.01 | 96.01 | 95.57 | 94.61 ± 0.28 |
| carpet | 93.26 | 57.70 | 98.76 | 97.31 | 97.91 | 57.95 | 97.59 | **99.70** | 98.50 ± 0.17 |
| grid | 93.57 | 76.02 | 99.08 | 84.04 | 97.24 | 93.07 | 94.57 | **100.0** | 92.67 ± 1.20 |
| hazelnut | **100.0** | 84.43 | 81.57 | 86.07 | **100.0** | 99.82 | **100.0** | 93.39 | 99.93 ± 0.05 |
| leather | 99.97 | 79.31 | **100.0** | 99.66 | **100.0** | 42.39 | 99.46 | 99.92 | **100.0** ± 0.00 |
| metal_nut | **99.76** | 45.06 | 94.53 | 96.92 | 99.61 | 67.20 | 93.06 | 99.34 | 98.88 ± 0.07 |
| pill | 90.73 | 44.65 | 87.53 | 88.52 | 94.35 | 54.66 | 92.06 | **99.08** | 96.17 ± 0.39 |
| screw | 88.30 | 89.38 | 66.00 | 75.24 | **98.26** | 94.57 | 93.69 | 98.09 | 95.20 ± 0.15 |
| tile | **100.0** | 90.15 | 95.42 | 95.49 | 98.67 | 97.37 | 92.97 | 99.85 | 99.99 ± 0.02 |
| toothbrush | 83.33 | 80.28 | 79.44 | 93.61 | **100.0** | 84.72 | **100.0** | **100.0** | **100.0** ± 0.00 |
| transistor | 91.50 | 88.37 | 94.42 | 92.29 | **100.0** | 83.29 | 80.54 | 96.57 | 97.65 ± 0.57 |
| wood | 98.33 | 90.96 | 97.54 | 98.33 | **99.30** | 53.16 | 98.77 | 98.13 | 98.81 ± 0.23 |
| zipper | 93.07 | 68.17 | 92.54 | 86.48 | **99.47** | 92.04 | 89.97 | 99.22 | 94.36 ± 0.13 |
| Mean | 94.47 | 75.26 | 90.72 | 91.25 | **98.75** | 79.65 | 93.54 | 98.23 | 97.65 ± 0.38 |

Table 8: Anomaly segmentation performance based on Pixel AUROC on MVTec dataset.

| Category | CFLOW (2021) | DRÆM (2021) | FastFlow (2021 | PaDiM (2021) | PatchCore (2022) | RD (2022) | DFR (2020) | EffAD (2024) | ULSAD (Ours) |
|---|---|---|---|---|---|---|---|---|---|
| bottle | **98.58** | 76.53 | 97.8 | 98.3 | 97.98 | 98.31 | 90.83 | 98.31 | 96.21 ± 2.21 |
| cable | 96.1 | 66.59 | 95.71 | 96.81 | 98.03 | 96.37 | 91.37 | **98.5** | 97.71 ± 0.06 |
| capsule | 98.71 | 86.96 | 98.37 | 98.67 | 98.77 | **98.96** | 98.46 | 98.33 | 98.95 ± 0.03 |
| carpet | 98.57 | 71.95 | 98.27 | 98.68 | 98.67 | 99.05 | 98.46 | 94.83 | **99.18** ± 0.06 |
| grid | 97.49 | 53.56 | 98.39 | 92.82 | 97.86 | **99.01** | 97.41 | 96.02 | 95.47 ± 1.09 |
| hazelnut | 98.64 | 84.66 | 94.79 | 97.85 | 98.43 | **98.91** | 98.53 | 96.15 | 98.81 ± 0.03 |
| leather | 99.42 | 63.32 | **99.62** | 99.30 | 98.87 | 99.17 | 99.33 | 97.5 | 98.68 ± 0.01 |
| metal_nut | 97.97 | 80.25 | 97.01 | 96.71 | **98.51** | 97.68 | 93.02 | 98.07 | 97.62 ± 0.03 |
| pill | 97.83 | 77.17 | 96.38 | 95.03 | 97.53 | 96.96 | 96.86 | **98.63** | 96.67 ± 0.09 |
| screw | 97.64 | 83.38 | 89.87 | 97.89 | 99.19 | **99.43** | 99.07 | 98.50 | 99.33 ± 0.01 |
| tile | **96.68** | 85.75 | 93.14 | 92.42 | 94.86 | 95.47 | 90.82 | 91.61 | 95.78 ± 0.05 |
| toothbrush | 98.16 | 90.70 | 97.50 | 98.83 | 98.67 | **98.99** | 98.49 | 96.0 | 98.42 ± 0.02 |
| transistor | 89.91 | 63.23 | 96.45 | 96.85 | 96.84 | 86.77 | 79.11 | 94.77 | **98.89** ± 0.05 |
| wood | 94.70 | 71.73 | **95.71** | 93.83 | 93.31 | 95.06 | 95.36 | 90.85 | 95.20 ± 0.31 |
| zipper | 97.08 | 69.31 | 97.62 | 97.82 | 98.06 | **98.54** | 96.85 | 96.21 | 97.24 ± 0.07 |
| Mean | 97.17 | 75.01 | 96.44 | 96.79 | **97.71** | 97.25 | 94.93 | 96.29 | 97.61 ± 0.64 |

Table 9: Anomaly segmentation performance based on Pixel AUPRO on MVTec dataset.

| Category | CFLOW (2021) | DRÆM (2021) | FastFlow (2021 | PaDiM (2021) | PatchCore (2022) | RD (2022) | DFR (2020) | EffAD (2024) | ULSAD (Ours) |
|---|---|---|---|---|---|---|---|---|---|
| bottle | 94.19 | 50.05 | 92.0 | 95.11 | 92.28 | **95.12** | 83.14 | 93.84 | 90.16 ± 3.24 |
| cable | 85.85 | 28.58 | 86.65 | 89.65 | 90.77 | 90.32 | 83.09 | **92.53** | 88.63 ± 0.48 |
| capsule | 90.47 | 81.11 | 90.15 | 92.62 | 92.4 | 93.93 | **96.33** | 91.09 | 93.77 ± 0.16 |
| carpet | 92.64 | 48.64 | 94.63 | 95.59 | 92.7 | **96.41** | 95.47 | 90.99 | 96.39 ± 0.24 |
| grid | 90.52 | 17.71 | 93.95 | 82.52 | 89.46 | **96.39** | 91.15 | 93.14 | 83.3 ± 3.96 |
| hazelnut | 96.12 | 76.19 | 93.92 | 92.95 | 94.44 | 96.92 | **97.17** | 83.25 | 94.87 ± 0.3 |
| leather | 98.39 | 52.1 | **99.06** | 97.91 | 96.33 | 97.97 | 98.34 | 97.32 | 97.44 ± 0.01 |
| metal_nut | 88.97 | 35.79 | 85.89 | 90.45 | 91.9 | **94.4** | 87.01 | 92.97 | 91.58 ± 0.19 |
| pill | 93.67 | 64.26 | 91.0 | 93.88 | 93.92 | 94.76 | 95.86 | **95.93** | 94.5 ± 0.08 |
| screw | 90.25 | 53.22 | 68.6 | 92.14 | 95.39 | **97.05** | 95.96 | 96.04 | 96.45 ± 0.1 |
| tile | **91.49** | 58.48 | 81.01 | 78.32 | 79.64 | 88.4 | 79.36 | 83.54 | 87.82 ± 0.15 |
| toothbrush | 81.05 | 54.02 | 80.62 | **93.52** | 86.48 | 92.23 | 92.93 | 88.61 | 86.28 ± 0.46 |
| transistor | 78.75 | 51.37 | 88.92 | 89.04 | **94.06** | 75.05 | 64.25 | 82.82 | 91.61 ± 0.68 |
| wood | 90.5 | 45.29 | **93.26** | 91.39 | 85.08 | 92.69 | 92.48 | 76.16 | 91.34 ± 0.29 |
| zipper | 89.3 | 28.98 | 92.12 | 92.48 | 92.43 | **95.18** | 88.74 | 93.48 | 90.89 ± 0.35 |
| Mean | 90.14 | 49.72 | 88.79 | 91.17 | 91.15 | **93.12** | 89.42 | 90.11 | 91.67 ± 1.36 |

Table 10: Anomaly detection performance based on Image AUROC on MVTecLOCO dataset.

| Category | CFLOW (2021) | DRÆM (2021) | FastFlow (2021 | PaDiM (2021) | PatchCore (2022) | RD (2022) | DFR (2020) | EffAD (2024) | ULSAD (Ours) |
|---|---|---|---|---|---|---|---|---|---|
| breakfast_box | 71.86 | 70.26 | 74.04 | 63.66 | **85.24** | 52.69 | 65.46 | 74.80 | 83.54 ± 0.23 |
| juice_bottle | 81.70 | 62.55 | 78.03 | 88.42 | 92.51 | 76.28 | 86.81 | 98.89 | **97.12** ± 0.10 |
| pushpins | 73.43 | 51.32 | 61.20 | 61.30 | 75.54 | 50.72 | 72.68 | 80.58 | **86.85** ± 0.94 |
| screw_bag | 65.48 | 59.39 | 68.04 | 60.14 | 69.90 | 65.15 | 63.55 | 67.42 | **70.71** ± 1.49 |
| splicing_connectors | 75.63 | 68.25 | 73.71 | 68.40 | **84.24** | 62.95 | 75.87 | 81.39 | 82.30 ± 0.72 |
| Mean | 73.62 | 62.35 | 71.00 | 68.38 | 81.49 | 61.56 | 72.87 | 80.62 | **84.1** ± 0.86 |

Table 11: Anomaly segmentation performance based on Pixel AUROC on MVTecLOCO dataset.

| Category | CFLOW (2021) | DRÆM (2021) | FastFlow (2021 | PaDiM (2021) | PatchCore (2022) | RD (2022) | DFR (2020) | EffAD (2024) | ULSAD (Ours) |
|---|---|---|---|---|---|---|---|---|---|
| breakfast_box | **89.6** | 63.61 | 82.73 | 87.35 | 88.53 | 85.78 | 76.25 | 80.76 | 89.14 ± 0.11 |
| juice_bottle | 91.37 | 80.71 | 86.33 | **91.99** | 90.54 | 90.41 | 87.06 | 88.40 | 89.07 ± 0.10 |
| pushpins | 70.66 | 54.74 | **82.94** | 40.72 | 67.67 | 41.42 | 29.42 | 59.96 | 75.64 ± 0.36 |
| screw_bag | 69.94 | 65.23 | 58.07 | 65.35 | 62.40 | 67.33 | 59.74 | 61.64 | **71.35** ± 0.12 |
| splicing_connectors | 63.40 | 54.16 | 67.69 | 71.20 | 69.71 | 57.82 | 56.14 | 61.02 | **75.10** ± 0.20 |
| Mean | 76.99 | 63.69 | 75.55 | 71.32 | 75.77 | 68.55 | 61.72 | 70.36 | **80.06** ± 0.20 |

Table 12: Anomaly segmentation performance based on Pixel AUPRO on MVTecLOCO dataset.

| Category | CFLOW (2021) | DRÆM (2021) | FastFlow (2021 | PaDiM (2021) | PatchCore (2022) | RD (2022) | DFR (2020) | EffAD (2024) | ULSAD (Ours) |
|---|---|---|---|---|---|---|---|---|---|
| breakfast_box | 67.27 | 36.11 | 63.8 | **74.28** | 73.08 | 69.67 | 63.56 | 58.44 | $71.36 \pm 0.38$ |
| juice_bottle | 80.75 | 51.51 | 77.90 | **88.78** | 85.42 | 84.95 | 82.88 | 86.51 | 87.72 $\pm 0.09$ |
| pushpins | 61.09 | 24.68 | 50.62 | 52.71 | 63.52 | 53.52 | 59.12 | 59.25 | **68.34** $\pm 0.55$ |
| screw_bag | 54.39 | 31.27 | 38.1 | 61.42 | 56.12 | 59.66 | **71.66** | 62.45 | 66.52 $\pm 0.33$ |
| splicing_connectors | 71.15 | 56.72 | 34.77 | 62.64 | 67.29 | 63.62 | 71.67 | 68.14 | **74.70** $\pm 0.25$ |
| Mean | 66.93 | 40.06 | 53.04 | 67.97 | 69.09 | 66.28 | 69.78 | 66.96 | **73.73** $\pm 0.35$ |

Table 13: Anomaly detection performance based on Image AUROC on MPDD dataset.

| Category | CFLOW (2021) | DRÆM (2021) | FastFlow (2021 | PaDiM (2021) | PatchCore (2022) | RD (2022) | DFR (2020) | EffAD (2024) | ULSAD (Ours) |
|---|---|---|---|---|---|---|---|---|---|
| tubes | **99.64** | 61.28 | 89.36 | 56.48 | 87.50 | 89.67 | 94.47 | 95.28 | $93.39 \pm 0.57$ |
| metal_plate | 97.42 | 80.05 | 86.92 | 42.69 | 99.72 | 91.87 | 68.31 | **100.0** | $93.81 \pm 0.28$ |
| connector | 99.52 | 83.33 | 52.38 | 86.07 | **100.0** | 93.10 | **100.0** | 50.00 | $96.00 \pm 0.82$ |
| bracket_white | 79.89 | 84.00 | 50.78 | 80.33 | 89.67 | 83.67 | 54.55 | 96.48 | **100.0** $\pm 0.00$ |
| bracket_black | **96.48** | 65.36 | 62.83 | 66.69 | 86.97 | 50.73 | 72.63 | 85.45 | 93.09 $\pm 0.32$ |
| bracket_brown | 49.70 | 70.81 | 47.89 | 78.66 | 95.78 | 68.70 | 88.54 | 85.32 | **98.08** $\pm 0.14$ |
| Mean | 87.11 | 74.14 | 65.03 | 68.48 | 93.27 | 79.62 | 79.75 | 85.42 | **95.73** $\pm 0.45$ |

Table 14: Anomaly segmentation performance based on Pixel AUROC on MPDD dataset.

| Category | CFLOW (2021) | DRÆM (2021) | FastFlow (2021 | PaDiM (2021) | PatchCore (2022) | RD (2022) | DFR (2020) | EffAD (2024) | ULSAD (Ours) |
|---|---|---|---|---|---|---|---|---|---|
| tubes | **99.15** | 76.87 | 98.44 | 91.35 | 98.45 | 99.08 | 98.62 | 98.98 | $98.55 \pm 0.10$ |
| metal_plate | **98.56** | 96.23 | 92.99 | 91.67 | 98.30 | 97.50 | 93.59 | 96.52 | $96.77 \pm 0.09$ |
| connector | 97.38 | 90.01 | 92.69 | 97.93 | 99.11 | 98.55 | 98.68 | 99.32 | **99.40** $\pm 0.18$ |
| bracket_white | 96.74 | 86.64 | 90.23 | 97.21 | 97.90 | 98.13 | 96.63 | 97.71 | **98.73** $\pm 0.1$ |
| bracket_black | 97.68 | 95.89 | 94.39 | 93.79 | 97.52 | 96.40 | **98.42** | 97.17 | $97.43 \pm 0.21$ |
| bracket_brown | 95.04 | 76.11 | 92.84 | 95.13 | 97.15 | 97.34 | **98.06** | 92.46 | $93.83 \pm 2.41$ |
| Mean | 97.42 | 86.96 | 93.60 | 94.51 | **98.07** | 97.83 | 97.33 | 97.03 | $97.45 \pm 0.99$ |

Table 15: Anomaly segmentation performance based on Pixel AUPRO on MPDD dataset.

| Category | CFLOW (2021) | DRÆM (2021) | FastFlow (2021 | PaDiM (2021) | PatchCore (2022) | RD (2022) | DFR (2020) | EffAD (2024) | ULSAD (Ours) |
|---|---|---|---|---|---|---|---|---|---|
| tubes | **96.76** | 44.84 | 94.85 | 71.53 | 93.83 | 95.98 | 95.20 | 96.27 | $94.33 \pm 0.33$ |
| metal_plate | 91.53 | 82.83 | 74.62 | 75.47 | **92.50** | 92.00 | 83.99 | 83.59 | $90.07 \pm 0.22$ |
| connector | 91.32 | 72.06 | 76.98 | 92.74 | 96.89 | 95.29 | 95.60 | 97.77 | **97.98** $\pm 0.60$ |
| bracket_white | 78.66 | 69.13 | 49.65 | 81.16 | 83.13 | 84.71 | 77.02 | 93.27 | **95.33** $\pm 0.37$ |
| bracket_black | 89.48 | 93.12 | 79.65 | 83.51 | 93.65 | 89.14 | **95.57** | 89.98 | $90.17 \pm 0.67$ |
| bracket_brown | 83.62 | 58.29 | 85.62 | 82.69 | 85.07 | 94.04 | **95.37** | 81.77 | $84.24 \pm 6.38$ |
| Mean | 88.56 | 70.04 | 76.89 | 81.18 | 90.84 | 91.86 | 90.46 | 90.44 | **92.02** $\pm 2.64$ |

Table 16: Anomaly detection performance based on Image AUROC on BTAD dataset.

| Category | CFLOW (2021) | DRÆM (2021) | FastFlow (2021 | PaDiM (2021) | PatchCore (2022) | RD (2022) | DFR (2020) | EffAD (2024) | ULSAD (Ours) |
|---|---|---|---|---|---|---|---|---|---|
| 01 | 98.64 | 80.17 | 94.46 | 99.51 | 98.09 | 92.23 | 99.51 | 94.15 | **100.0** ± 0.00 |
| 02 | 82.12 | 65.23 | 84.27 | 82.17 | 81.73 | 61.73 | 85.68 | 75.42 | **88.5** ± 0.78 |
| 03 | 99.95 | 74.87 | 96.3 | 97.92 | **100.0** | 97.65 | 98.62 | 95.22 | **100.0** ± 0.00 |
| Mean | 93.57 | 73.42 | 91.68 | 93.20 | 93.27 | 83.87 | 94.60 | 88.26 | **96.17** ± 0.45 |

Table 17: Anomaly segmentation performance based on Pixel AUROC on BTAD dataset.

| Category | CFLOW (2021) | DRÆM (2021) | FastFlow (2021 | PaDiM (2021) | PatchCore (2022) | RD (2022) | DFR (2020) | EffAD (2024) | ULSAD (Ours) |
|---|---|---|---|---|---|---|---|---|---|
| 01 | 95.44 | 59.11 | 93.05 | 96.54 | 95.94 | **96.98** | 96.93 | 64.59 | 95.86 ± 0.03 |
| 02 | 94.81 | 69.29 | 96.16 | 95.11 | 95.18 | **96.83** | 96.77 | 85.67 | 94.76 ± 0.88 |
| 03 | 99.55 | 48.73 | 99.25 | 99.56 | 99.44 | **99.74** | 99.12 | 96.12 | 99.55 ± 0.02 |
| Mean | 96.60 | 59.04 | 96.15 | 97.07 | 96.85 | **97.85** | 97.62 | 82.13 | 96.73 ± 0.51 |

Table 18: Anomaly segmentation performance based on AUPRO on BTAD dataset.

| Category | CFLOW (2021) | DRÆM (2021) | FastFlow (2021 | PaDiM (2021) | PatchCore (2022) | RD (2022) | DFR (2020) | EffAD (2024) | ULSAD (Ours) |
|---|---|---|---|---|---|---|---|---|---|
| 01 | 66.79 | 21.57 | 60.83 | 75.76 | 64.34 | 79.45 | **83.77** | 29.75 | 72.88 ± 0.12 |
| 02 | 54.32 | 27.64 | **67.98** | 59.19 | 52.36 | 66.05 | 65.58 | 44.37 | 55.16 ± 6.83 |
| 03 | 98.21 | 18.24 | 96.99 | 98.45 | 97.76 | **98.92** | 27.83 | 88.98 | 98.18 ± 0.08 |
| Mean | 73.11 | 22.48 | 75.27 | 77.80 | 71.48 | **81.47** | 59.06 | 54.37 | 75.41 ± 3.95 |

Table 19: Anomaly detection performance based on Image AUROC on VisA dataset.

| Category | CFLOW (2021) | DRÆM (2021) | FastFlow (2021 | PaDiM (2021) | PatchCore (2022) | RD (2022) | DFR (2020) | EffAD (2024) | ULSAD (Ours) |
|---|---|---|---|---|---|---|---|---|---|
| candle | 94.38 | 79.43 | 93.18 | 86.19 | **98.59** | 85.54 | 89.65 | 80.52 | 87.11 ± 0.29 |
| capsules | 69.9 | 72.77 | 81.05 | 61.72 | 69.92 | **87.37** | 76.75 | 63.73 | 79.61 ± 0.72 |
| cashew | 94.7 | 95.5 | 87.78 | 90.94 | **96.29** | 85.38 | 93.80 | 96.11 | 94.72 ± 0.16 |
| chewinggum | 99.02 | 83.68 | 95.18 | 98.20 | 99.29 | 81.92 | 99.22 | 98.27 | **99.49** ± 0.12 |
| fryum | 92.98 | 70.46 | 92.60 | 85.06 | 93.5 | 77.94 | **96.58** | 95.70 | 95.86 ± 0.14 |
| macaroni1 | 92.72 | 72.8 | 82.48 | 78.62 | 91.50 | 82.06 | 95.14 | **95.23** | 90.66 ± 0.76 |
| macaroni2 | 63.44 | 47.85 | 69.75 | 70.05 | 71.36 | 81.75 | **86.25** | 83.82 | 82.84 ± 1.05 |
| pcb1 | 91.06 | 72.27 | 88.07 | 87.59 | 95.08 | 92.60 | **97.57** | 93.78 | 92.92 ± 0.11 |
| pcb2 | 79.95 | 91.17 | 86.47 | 83.20 | 92.46 | 87.57 | 91.55 | **94.95** | 93.67 ± 0.18 |
| pcb3 | 82.23 | 81.29 | 81.47 | 72.79 | 92.46 | 90.87 | **97.27** | 95.92 | 93.62 ± 0.16 |
| pcb4 | 96.29 | 90.44 | 95.68 | 95.67 | 99.20 | 96.17 | 97.62 | 97.89 | **99.43** ± 0.03 |
| pipe_fryum | 96.54 | 75.32 | 96.16 | 89.28 | 98.07 | 85.68 | 98.36 | 98.59 | **99.61** ± 0.11 |
| Mean | 87.77 | 77.75 | 87.49 | 83.28 | 91.48 | 86.24 | 85.18 | 91.21 | **92.46** ± 0.45 |

Table 20: Anomaly segmentation performance based on Pixel AUROC on VisA dataset.

| Category | CFLOW (2021) | DRÆM (2021) | FastFlow (2021 | PaDiM (2021) | PatchCore (2022) | RD (2022) | DFR (2020) | EffAD (2024) | ULSAD (Ours) |
|---|---|---|---|---|---|---|---|---|---|
| candle | 98.75 | 83.1 | 97.24 | 97.68 | 98.92 | **99.11** | 98.41 | 89.93 | 97.77 ± 0.03 |
| capsules | 96.88 | 62.39 | 97.13 | 90.60 | 97.62 | **99.56** | 99.13 | 96.93 | 98.31 ± 0.31 |
| cashew | 99.25 | 74.17 | 98.57 | 97.45 | 98.88 | 97.23 | 95.63 | 98.85 | **99.49** ± 0.02 |
| chewinggum | 99.02 | 84.11 | 98.83 | 98.82 | 98.72 | **99.37** | 99.16 | 98.69 | 98.10 ± 0.41 |
| fryum | 97.08 | 85.7 | 93.20 | 96.20 | 94.30 | 96.33 | 95.45 | 96.52 | **97.38** ± 0.19 |
| macaroni1 | 98.71 | 63.95 | 98.60 | 97.85 | 98.13 | 99.48 | **99.73** | 99.59 | 99.00 ± 0.13 |
| macaroni2 | 97.35 | 79.02 | 94.65 | 95.40 | 96.79 | 99.33 | **99.43** | 98.84 | 98.20 ± 0.28 |
| pcb1 | 99.05 | 27.98 | 99.29 | 98.67 | 99.47 | **99.65** | 99.30 | 98.98 | 99.61 ± 0.01 |
| pcb2 | 96.40 | 59.49 | 97.12 | 98.12 | 97.72 | 98.28 | 96.13 | **98.37** | 98.03 ± 0.09 |
| pcb3 | 97.23 | 76.43 | 97.04 | 98.06 | 98.13 | **98.98** | 97.99 | 98.91 | 98.45 ± 0.05 |
| pcb4 | 97.97 | 83.42 | 97.51 | 97.00 | 97.83 | **98.29** | 96.58 | 95.49 | 95.22 ± 0.27 |
| pipe_fryum | 98.79 | 75.99 | 98.72 | 99.19 | 98.68 | 98.6 | 97.97 | 98.99 | **99.29** ± 0.03 |
| Mean | 98.04 | 71.31 | 97.32 | 97.09 | 97.93 | **98.68** | 97.90 | 97.51 | 98.24 ± 0.20 |

Table 21: Anomaly segmentation performance based on AUPRO on VisA dataset.

| Category | CFLOW (2021) | DRÆM (2021) | FastFlow (2021 | PaDiM (2021) | PatchCore (2022) | RD (2022) | DFR (2020) | EffAD (2024) | ULSAD (Ours) |
|---|---|---|---|---|---|---|---|---|---|
| candle | 92.7 | 80.29 | 91.65 | 92.77 | 94.08 | 95.30 | **95.56** | 77.31 | 92.49 ± 0.15 |
| capsules | 74.64 | 34.4 | 81.8 | 48.42 | 68.88 | **92.20** | 92.09 | 83.8 | 82.76 ± 1.18 |
| cashew | **93.0** | 48.33 | 85.54 | 82.36 | 88.01 | 91.81 | 90.51 | 91.57 | 91.85 ± 1.15 |
| chewinggum | **89.58** | 62.66 | 84.69 | 84.33 | 83.86 | 88.57 | 85.52 | 74.87 | 84.34 ± 1.0 |
| fryum | 85.62 | 71.94 | 72.39 | 75.54 | 78.25 | 84.8 | **92.08** | 82.93 | 85.47 ± 0.66 |
| macaroni1 | 89.46 | 63.37 | 91.89 | 88.55 | 91.74 | 95.53 | **97.59** | 96.06 | 92.8 ± 0.74 |
| macaroni2 | 78.74 | 56.69 | 71.94 | 75.76 | 87.49 | 94.01 | **94.23** | 89.74 | 88.29 ± 1.89 |
| pcb1 | 87.24 | 27.43 | 85.89 | 86.39 | 89.07 | **95.0** | 93.55 | 90.53 | 90.22 ± 0.28 |
| pcb2 | 77.83 | 33.99 | 77.99 | 83.68 | 83.00 | 89.17 | 87.26 | **90.43** | 84.53 ± 0.53 |
| pcb3 | 75.03 | 71.9 | 71.33 | 81.37 | 79.69 | 90.89 | **92.48** | 92.08 | 85.86 ± 0.38 |
| pcb4 | 86.53 | 73.26 | 83.41 | 82.47 | 84.91 | **89.17** | 84.38 | 75.25 | 73.37 ± 0.81 |
| pipe_fryum | 93.14 | 31.86 | 81.89 | 87.99 | 92.42 | 94.76 | **95.46** | 68.77 | 93.49 ± 0.08 |
| Mean | 85.29 | 54.68 | 81.70 | 80.80 | 85.12 | **91.77** | 91.72 | 84.45 | 87.12 ± 0.89 |

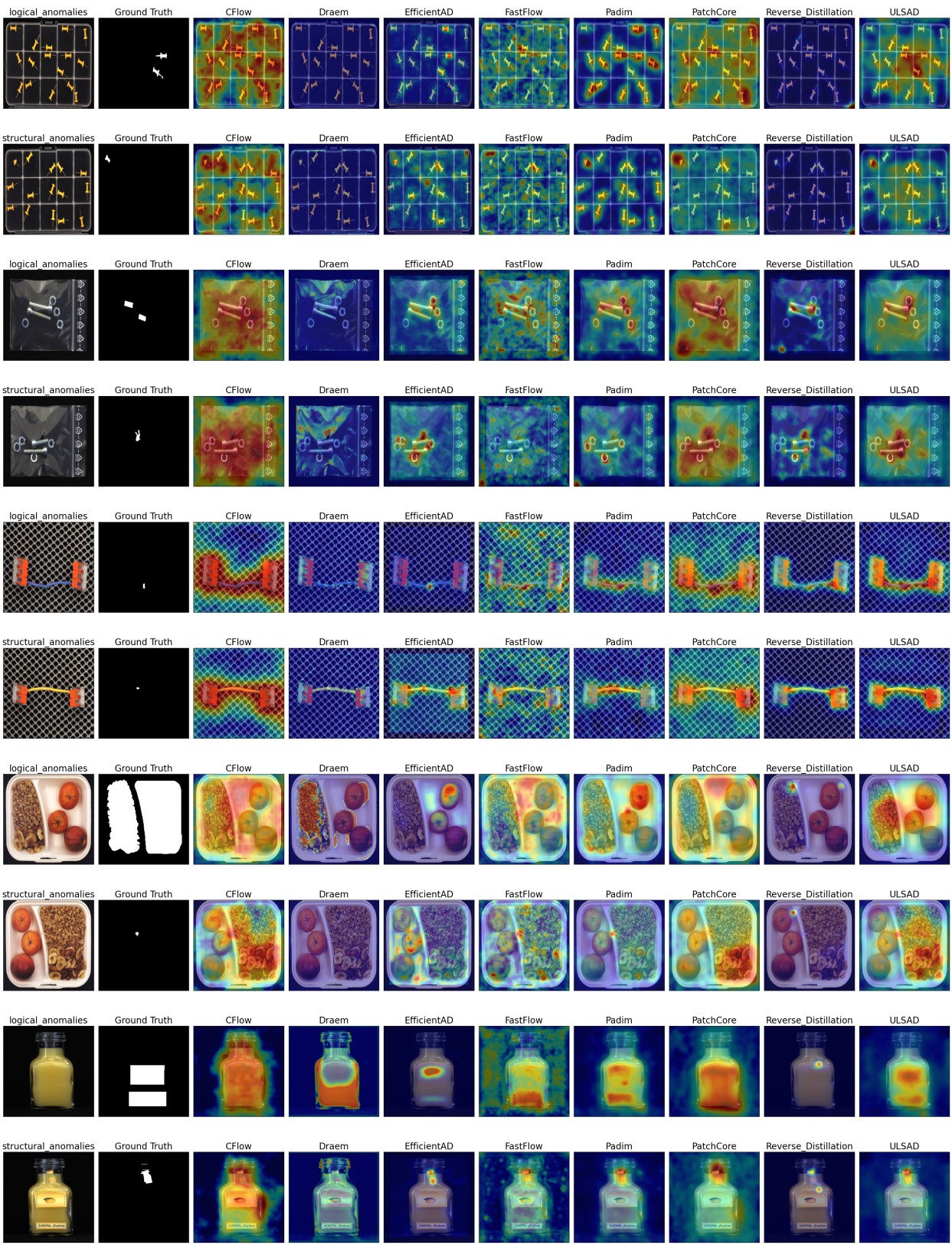

Figure 9: Visualization of anomaly maps on anomalous images from MVTecLOCO dataset.

## B.1 Performance on MVTecLOCO: Logical and Structural AD

In Tables 22, 23, 24, 25, 26, and 27, we report the anomaly detection and localization results on MVTec LOCO separately for structural and logical anomalies. It can be observed that although PatchCore performs slightly better than ULSAD on structural anomalies, ULSAD delivers competitive results on logical anomalies. Moreover, as discussed previously in Section 5, the improvement in performance with PatchCore comes with three times the memory requirement. Therefore, we consider ULSAD to be an efficient and effective approach for the detection and localization of both logical and structural anomalies.

Table 22: Anomaly detection performance based on Image AUROC on **structural anomalies** of MVTecLOCO dataset.

| Category | CFLOW (2021) | DRÆM (2021) | FastFlow (2021 | PaDiM (2021) | PatchCore (2022) | RD (2022) | DFR (2020) | EffAD (2024) | ULSAD (Ours) |
|---|---|---|---|---|---|---|---|---|---|
| breakfast_box | 62.3 | 75.37 | 71.67 | 64.17 | **84.3** | 48.74 | 58.67 | 69.04 | $\underline{81.94} \pm 0.74$ |
| juice_bottle | 73.45 | 52.18 | 76.52 | 86.17 | $\underline{96.47}$ | 78.29 | 62.75 | **99.71** | $95.53 \pm 0.45$ |
| pushpins | 71.03 | 66.51 | 60.17 | 72.4 | 77.42 | 50.17 | 40.06 | **92.07** | $\underline{85.17} \pm 0.65$ |
| screw_bag | 78.0 | 69.78 | 75.07 | 67.7 | **86.79** | 80.35 | 57.76 | 82.2 | $\underline{83.01} \pm 1.56$ |
| splicing_connectors | 74.35 | 80.33 | 70.31 | 66.85 | $\underline{88.67}$ | 63.04 | 55.84 | **90.2** | $80.59 \pm 0.35$ |
| Mean | 71.83 | 68.83 | 70.75 | 71.46 | **86.73** | 64.12 | 55.02 | $\underline{86.64}$ | $85.25 \pm 0.86$ |

Table 23: Anomaly segmentation performance based on Pixel AUROC on **structural anomalies** of MVTecLOCO dataset.

| Category | CFLOW (2021) | DRÆM (2021) | FastFlow (2021 | PaDiM (2021) | PatchCore (2022) | RD (2022) | DFR (2020) | EffAD (2024) | ULSAD (Ours) |
|---|---|---|---|---|---|---|---|---|---|
| breakfast_box | $\underline{94.72}$ | 61.6 | 84.86 | 88.41 | 94.31 | 91.75 | 82.79 | 67.36 | **95.35** $\pm 0.16$ |
| juice_bottle | 88.81 | 83.75 | 85.67 | 90.73 | $\underline{95.98}$ | 89.57 | 80.74 | **97.24** | $89.67 \pm 0.23$ |
| pushpins | $\underline{91.4}$ | 46.14 | 85.54 | 90.82 | **95.16** | 87.83 | 48.52 | 89.48 | $88.38 \pm 0.22$ |
| screw_bag | 95.37 | 72.27 | 91.74 | 94.97 | 97.45 | **97.78** | 66.67 | 97.37 | $\underline{97.64} \pm 0.24$ |
| splicing_connectors | 98.3 | 93.03 | 95.79 | 96.25 | **98.82** | 97.25 | 91.82 | $\underline{98.55}$ | $98.3 \pm 0.1$ |
| Mean | 93.72 | 71.36 | 88.72 | 92.24 | **96.34** | 92.84 | 74.12 | 90.0 | $\underline{93.87} \pm 0.2$ |

Table 24: Anomaly segmentation performance based on Pixel AUPRO on **structural anomalies** of MVTecLOCO dataset.

| Category | CFLOW (2021) | DRÆM (2021) | FastFlow (2021 | PaDiM (2021) | PatchCore (2022) | RD (2022) | DFR (2020) | EffAD (2024) | ULSAD (Ours) |
|---|---|---|---|---|---|---|---|---|---|
| breakfast_box | 70.0 | 41.92 | 71.29 | **81.97** | $\underline{78.71}$ | 76.67 | 26.67 | 65.87 | $75.43 \pm 0.67$ |
| juice_bottle | 80.56 | 50.44 | 87.61 | $\underline{92.64}$ | **95.1** | 90.81 | 61.67 | 92.08 | $90.1 \pm 0.31$ |
| pushpins | 68.34 | 20.04 | 61.72 | 70.01 | $\underline{75.88}$ | 62.63 | 45.30 | **78.62** | $68.34 \pm 0.84$ |
| screw_bag | 82.07 | 39.41 | 73.69 | 85.1 | 88.92 | **93.77** | 53.95 | 91.38 | $\underline{91.98} \pm 0.73$ |
| splicing_connectors | 87.75 | 68.61 | 67.8 | 82.35 | $\underline{91.69}$ | 83.57 | 63.44 | **94.19** | $90.84 \pm 0.35$ |
| Mean | 77.74 | 44.08 | 72.42 | 82.41 | **86.06** | 81.49 | 50.21 | $\underline{84.43}$ | $83.34 \pm 0.62$ |

Table 25: Anomaly detection performance based on Image AUROC on **logical anomalies** of MVTecLOCO dataset.

| Category | CFLOW (2021) | DRÆM (2021) | FastFlow (2021 | PaDiM (2021) | PatchCore (2022) | RD (2022) | DFR (2020) | EffAD (2024) | ULSAD (Ours) |
|---|---|---|---|---|---|---|---|---|---|
| breakfast_box | 76.86 | 68.71 | 75.81 | 62.0 | 83.43 | 55.27 | 61.55 | **83.54** | 83.33 ± 1.63 |
| juice_bottle | 80.52 | 70.09 | 80.89 | 92.94 | 94.61 | 76.4 | 77.07 | **99.12** | 98.82 ± 0.14 |
| pushpins | 71.15 | 37.9 | 58.81 | 50.72 | 74.53 | 52.98 | 60.27 | 72.0 | **85.23** ± 0.72 |
| screw_bag | 60.72 | 52.34 | 64.01 | 55.29 | 59.36 | 55.09 | 63.64 | 58.8 | **66.33** ± 0.9 |
| splicing_connectors | 67.5 | 56.15 | 74.54 | 69.43 | 80.27 | 61.74 | 53.10 | 75.64 | **86.27** ± 1.33 |
| Mean | 71.35 | 57.04 | 70.81 | 66.08 | 78.44 | 60.3 | 63.13 | 77.82 | **84.0** ± 1.07 |

Table 26: Anomaly segmentation performance based on Pixel AUROC on **logical anomalies** of MVTecLOCO dataset.

| Category | CFLOW (2021) | DRÆM (2021) | FastFlow (2021 | PaDiM (2021) | PatchCore (2022) | RD (2022) | DFR (2020) | EffAD (2024) | ULSAD (Ours) |
|---|---|---|---|---|---|---|---|---|---|
| breakfast_box | 90.43 | 65.5 | 85.35 | 89.23 | 90.32 | 87.35 | 74.55 | 85.58 | **91.41** ± 0.1 |
| juice_bottle | 92.56 | 80.49 | 90.23 | **95.29** | 93.97 | 92.97 | 86.95 | 91.98 | 93.74 ± 0.08 |
| pushpins | 70.38 | 56.06 | **82.91** | 41.95 | 69.39 | 41.86 | 67.93 | 61.12 | 75.96 ± 1.11 |
| screw_bag | 67.75 | 65.43 | 58.93 | 66.02 | 63.81 | 68.4 | **75.75** | 61.85 | 72.3 ± 0.41 |
| splicing_connectors | 60.8 | 52.1 | 66.82 | 69.83 | 68.06 | 55.5 | 57.67 | 59.01 | **74.19** ± 0.19 |
| Mean | 76.38 | 63.92 | 76.85 | 72.46 | 77.11 | 69.22 | 72.57 | 71.91 | **81.52** ± 0.54 |

Table 27: Anomaly segmentation performance based on Pixel AUPRO on **logical anomalies** of MVTecLOCO dataset.

| Category | CFLOW (2021) | DRÆM (2021) | FastFlow (2021 | PaDiM (2021) | PatchCore (2022) | RD (2022) | DFR (2020) | EffAD (2024) | ULSAD (Ours) |
|---|---|---|---|---|---|---|---|---|---|
| breakfast_box | 68.79 | 32.23 | 64.39 | 69.74 | 72.27 | 68.77 | 43.60 | 52.19 | **73.97** ± 0.8 |
| juice_bottle | 79.67 | 52.72 | 77.9 | 91.69 | 87.88 | 84.22 | 62.87 | 87.82 | **91.36** ± 0.12 |
| pushpins | 59.07 | 26.21 | 47.71 | 51.93 | 63.47 | 53.84 | 41.80 | 58.36 | **68.78** ± 0.98 |
| screw_bag | 50.7 | 26.98 | 25.58 | 53.92 | 46.8 | 48.72 | 54.59 | 52.8 | **61.48** ± 0.45 |
| splicing_connectors | 65.86 | 53.71 | 26.66 | 57.66 | 62.21 | 58.85 | 34.72 | 62.7 | **72.66** ± 0.27 |
| Mean | 64.82 | 38.37 | 48.45 | 64.99 | 66.53 | 62.88 | 47.52 | 62.77 | **73.65** ± 0.61 |

## C   Extended Ablations

In this section, we provide additional ablations on the local branch in Table 28 and the total architecture in Table 29. Lastly, in Table 30 we provide the per-category results for the ablation on the pre-trained backbone which is summarized in Figure 7.

Table 28: Abalations for local branch. Style: I-AUROC | P-AUROC | P-AUPRO.

| category | $\lambda_l = 0$ | $\lambda_l = 0.01$ | $\lambda_l = 0.5$ | $\lambda_l = 0.9$ | $\lambda_l = 1.0$ |
|---|---|---|---|---|---|
| breakfast_box | 78.64 | 88.28 | 74.22 | 79.2 | **88.51** | **74.35** | 77.86 | 87.79 | 71.27 | 77.95 | 86.89 | 67.06 | **79.44** | 86.96 | 65.36 |
| juice_bottle | **97.82** | 92.14 | 89.24 | 97.76 | **92.23** | 89.38 | 97.56 | 88.78 | 88.16 | 97.36 | 84.39 | 84.63 | 97.08 | 83.61 | 83.47 |
| pushpins | 72.4 | 69.81 | 65.69 | 72.77 | 69.84 | 65.68 | **79.92** | **74.49** | **69.03** | 76.98 | 74.35 | 63.17 | 76.53 | 73.69 | 65.18 |
| screw_bag | 66.42 | 66.6 | 64.39 | 67.18 | 68.47 | 65.92 | 68.06 | 69.13 | **66.22** | 67.56 | **69.33** | 63.61 | 66.34 | 69.31 | 62.09 |
| splicing_connectors | **73.05** | 59.04 | 73.3 | 72.84 | 59.15 | 73.29 | 72.29 | 62.66 | 72.39 | 72.79 | 64.33 | 70.74 | 72.36 | **64.5** | 70.57 |
| Mean | 77.67 | 75.17 | 73.37 | 77.95 | 75.64 | **73.72** | **79.14** | **76.57** | 73.41 | 78.53 | 75.86 | 69.84 | 78.35 | 75.61 | 69.33 |

Table 29: Abalations for total architecture. Style: I-AUROC | P-AUROC | P-AUPRO.

| category | $\mathcal{L}_{pg}^{d}$ | $\mathcal{L}_{pg}^{d}$;  $\mathcal{L}_{lg}$ | $\mathcal{L}_{pg}$ | $\mathcal{L}_{pg}$;  $\mathcal{L}_{lg}$ |
|---|---|---|---|---|
| | $\lambda_l = \lambda_g = 0.0$ | | | |
| breakfast_box | 77.29 | **90.41** | 77.16 | 82.82 | 89.85 | 76.92 | 66.01 | 87.36 | 67.5 | **85.08** | 90.19 | 75.36 |
| juice_bottle | 96.48 | **92.01** | 88.82 | **97.93** | 91.82 | **89.26** | 91.2 | 91.98 | 85.38 | 97.29 | 92.0 | 89.18 |
| pushpins | 70.89 | 80.89 | 70.67 | **78.66** | **88.09** | **79.11** | 74.67 | 77.37 | 58.75 | 74.61 | 85.86 | 76.33 |
| screw_bag | 65.48 | 65.87 | 64.04 | 63.02 | 68.13 | **65.51** | 61.14 | 59.51 | 57.75 | **66.93** | **68.67** | 65.35 |
| splicing_connectors | 78.29 | 69.68 | 75.61 | **84.55** | 72.71 | **76.54** | 65.31 | 53.4 | 66.74 | 81.5 | **73.11** | 76.01 |
| Mean | 77.69 | 79.77 | 75.26 | **81.4** | **82.12** | **77.47** | 71.67 | 73.92 | 67.22 | 81.08 | 81.97 | 76.45 |
| | $\lambda_l = \lambda_g = 0.5$ | | | |
| breakfast_box | 79.22 | **90.87** | 78.32 | 82.45 | 88.49 | 71.03 | 71.36 | 87.64 | 67.38 | **83.36** | 89.34 | 72.36 |
| juice_bottle | 96.3 | 91.17 | **88.89** | **98.08** | 87.06 | 88.05 | 91.14 | **91.84** | 85.92 | 97.46 | 88.81 | 87.71 |
| pushpins | 79.86 | 84.89 | 77.97 | 82.46 | **87.62** | **80.49** | 78.64 | 81.14 | 65.12 | **88.07** | 74.22 | 66.45 |
| screw_bag | 66.58 | 68.83 | 66.11 | 65.11 | 70.04 | 62.81 | 62.09 | 67.32 | 62.92 | **70.6** | **71.58** | **67.01** |
| splicing_connectors | 80.53 | 73.47 | **75.44** | **82.85** | 73.03 | 75.13 | 69.33 | 55.02 | 63.69 | 81.27 | **74.94** | 74.89 |
| Mean | 80.5 | **81.85** | **77.35** | 82.19 | 81.25 | 75.5 | 74.51 | 76.59 | 69.01 | **84.15** | 79.78 | 73.68 |

Table 30: Abalations for backbone on MvTec-LOCO. Style: I-AUROC | P-AUROC | P-AUPRO.

| Class | ResNet50 | ResNet152 | Wide-ResNet50-2 | Wide-ResNet100-2 |
|---|---|---|---|---|
| breakfast_box | 82.41 | 89.74 | 73.09 | **85.11** | **91.15** | 72.0 | 84.46 | 89.18 | 72.21 | 82.37 | 89.25 | **74.02** |
| juice_bottle | 96.9 | 92.23 | 89.38 | 97.64 | 91.66 | 89.78 | 97.11 | 88.9 | 87.94 | **98.74** | **92.87** | **91.07** |
| pushpins | 81.79 | **79.49** | **75.15** | 73.28 | 76.49 | 63.71 | **85.48** | 75.46 | 67.82 | 82.48 | 76.55 | 70.67 |
| screw_bag | 66.46 | 69.14 | **67.01** | 68.06 | 68.63 | 66.97 | 71.14 | **71.57** | 66.82 | **73.1** | 68.99 | 66.88 |
| splicing_connectors | 80.88 | 72.76 | 74.39 | 83.53 | 76.31 | 77.24 | 82.59 | 75.21 | 75.05 | **84.71** | **79.95** | **78.65** |
| Mean | 81.79 | 79.49 | 75.15 | 81.52 | 80.85 | 73.94 | 84.16 | 80.06 | 73.97 | **84.28** | **81.52** | **76.26** |

