# OpenReview forum: "Revisiting Deep Feature Reconstruction for Logical and Structural Industrial Anomaly Detection"
_TMLR — Accepted by TMLR_

### Review · Reviewer_UCFP · 2024-05-23

**Summary Of Contributions:**

This paper tackles Industrial Anomaly Detection (IAD) and handles structural anomalies (Bergmann et al., 2019) and logical anomalies (Bergmann et al., 2022).
Unified framework for Logical and Structural Anomaly Detection (ULSAD) is proposed based on Deep Feature Reconstruction (DFR) (Yang 2020). Two modifications from DFR are proposed.
- The combination of L2 distance and cosine distance between feature and reconstructed features is proposed for improving structural AD.
- Attention-based loss based on global autoencoder-like network, is proposed for improving logical AD.

The experiments are conducted on 5 widely used datasets and compared with 8 SOTA methods.

**Audience:**

Yes

**Claims And Evidence:**

No

**Requested Changes:**

-	Clarify the novelty of ULSAD.
-	Compare the model size and computational resources with SOTA methods to claim the computational efficiency.
-	Explain why adding $L_{pg}$ in the 3-th and 8-th rows of the results of Table 3 deceased the performance.
-	Sec.3.3, proposes to use the cross-attention of Eq.(4) instead of the self-attention of Eq.(3). Compare these variants.
-	Explain what the difference of the last two rows of Table 3 is.

**Strengths And Weaknesses:**

Strengths.
- ULSAD handles both structural and logical anomalies in the same frame work.

Weakness

- Novelty is limited or not clear for me.

The combination of L2 distance and cosine distance is the same as (Sahel et al., 2021) and there is no explanation why this combination improves performance.

The concept of attention-based loss seems to be similar to (Zhang et al, 2023)

-	On Table 1, the performance of ULSAD is lower than DFR.
-	The average segmentation performances of ULSAD on Table 2 are lower than state-of-the-art on several datasets.

---

> ### Author Response · Authors · 2024-08-08
> **Response to Reviewer UCFP (1/2)**
>
> We thank the reviewer for their insightful and valuable feedback. We address the reviewer's concerns in the points below.
>
> **[W1, RC1]** Novelty is limited or not clear for me. Clarify the novelty of ULSAD.
>
> We acknowledge that the combination of $\ell_2$ and cosine distance is similar to Salehi et al., 2021. However, in Salehi et al., 2021, a distillation-based approach was proposed to detect structural anomalies. We not only apply it for the detection of structural anomalies in the local branch but also to compute the differences between the attention maps in the global branch, which allows ULSAD to detect logical anomalies.
>
> As discussed in our introduction, the $\ell_2$ distance is not informative in high-dimensional spaces (Aggarwal et al., 2001), the incorporation of cosine distance with $\ell_2$ distance allows us to consider both the difference in value and direction of the feature vectors. Thus, this combination improves the performance of ULSAD. We have updated our paper to highlight this in Section 3.2.
>
> We acknowledge that the concept of using cosine similarity between each vector with every other vector in the feature map is similar to Zhang et al., 2023 as both are aimed at the detection of logical anomalies that violate the spatial layout constraints in normal images. However, there are several key differences between the two methods. Firstly, in the dual-student knowledge distillation (DSKD) method (Zhang et al., 2023), the teacher network takes an image as an input, and the student network takes the output of the teacher network as an input. Then, similar to Reverse Distillation (Deng \& Li, 2022), the loss is computed between the feature maps at each intermediate stage. On the contrary, in our method, ULSAD, both the global branch and the pre-trained network take the image as the input. Then, the loss is computed between the output of the global network and the pre-trained network. Secondly, in DSKD, after computing the affinity scores, the authors propose to use the KL divergence for computing the loss, while in ULSAD, we compute $\ell_2$ and cosine distances between the self and cross-attention maps.
>
> While we use existing methods, our contribution is the proposed combination of cosine similarity with $\ell_2$ distance and the use of attention maps in ULSAD. Furthermore, our goal in this paper is to improve DFR to match the performance of the SOTA methods while maintaining its computational and memory efficiency. Our experiments demonstrate that ULSAD can detect logical anomalies, a capability the original DFR lacked, and achieve competitive results in detecting and localizing structural anomalies compared to SOTA methods.
>
> **[W2]** On Table 1, the performance of ULSAD is lower than DFR.
>
> Thank you for pointing that out. Table 1 is a reduced version of Table 19 (Table 18 in the previously submitted version). We discovered an error in the values for the VisA dataset (last column) in the main text's reduced table. The full Table 18 provided in the Appendix (Table 19 in the current version) was free of errors. We have updated the values in Table 1 accordingly and ensured the accuracy of every reported value through cross-checking. We apologize for any confusion caused by this mistake and appreciate your understanding.
>
> As expected, we can see that ULSAD is better than DFR across all the benchmarking datasets for anomaly detection.
>
> **[W3]** The average segmentation performances of ULSAD on Table 2 are lower than SOTA.
>
> While we agree with the reviewer that ULSAD's average segmentation performance in Table 2 is lower than SOTA's on some datasets, the difference is not significant. The only significant difference that we can observe is in the AUPRO score on the BTAD, MVTec, and VisA datasets. Nonetheless, while the SOTA method provides slightly better performance in the localization of structural anomalies, ULSAD provides similar performance across both logical and structural anomalies. For the MVTecLOCO dataset, we added additional results in Appendix B.1 of our updated paper, comparing the performance separately for logical and structural anomalies. Furthermore, if we compare the performance, memory and throughput of all the methods, as shown in Table 4 of our updated paper, we can observe that ULSAD provides a better trade-off than the SOTA methods.
>
> **[RC2]** Compare the model size and computational resources.
>
> Please refer to our general comment on memory and computational complexity.
>
> **References:**
> - Salehi et al., Multiresolution knowledge distillation for anomaly detection, CVPR 2021.
> - Aggarwal et al., On the surprising behavior of distance metrics in high dimensional space, ICDT 2001.
> - Zhang et al., Contextual Affinity Distillation for Image Anomaly Detection, arxiv 2023.
> - Deng \& Li, Anomaly Detection via Reverse Distillation From One-Class Embedding, CVPR 2022.

---

> ### Author Response · Authors · 2024-08-08
> **Response to Reviewer UCFP (2/2)**
>
> **[RC3]** Explain why adding $L_{pg}$ in the 3-th and 8-th rows of the results of Table 3 deceased the performance.
>
> The 3rd and 8th rows of Table 3 show the performance of ULSAD when only using $L_{pg}$. The global loss $L_{pg}$ is incorporated to learn the spatial relationships among the objects in the normal images for detecting logical anomalies. So, when used in isolation, it limits the performance of ULSAD to detecting only spatial anomalies and cannot capture localized structural anomalies. This is observed as a decrease in the performance in Table 3. Furthermore, as we highlighted in the paper, $L_{pg}$  suffers from false positives when dealing with sharp edges or heavily textured backgrounds, which also contributes to the decrease in performance. We have also added the explanation in Section 4.3 of our updated draft.
>
> **[RC4]** Sec.3.3, proposes to use the cross-attention of Eq.(4) instead of the self-attention of Eq.(3). Compare these variants.
>
> In Section 3.3, we proposed to use cross-attention instead of self-attention to reduce the computational complexity of the optimization problem. A similar observation is made in Zhang et al., 2023 [Section 3.4]. Therefore, we did not provide results using self-attention in the paper. However, following the reviewer's comment, we provide the comparison in the Table (https://ibb.co/H2pXQnx). We can see that both the anomaly detection and localization performance are lower when using self-attention compared to when cross-attention is used in computing the loss.
>
> **[RC5]** Explain what the difference of the last two rows of Table 3 is.
>
> Thank you for pointing it out. The last two rows of Table 3 are for the same configuration. While the second-last row is the result from one of five runs, the last row is the aggregated result over all five runs. We have removed the second last row from the table in our updated draft to avoid further confusion.
>
> **Reference:**
> - Zhang et al., Contextual Affinity Distillation for Image Anomaly Detection, arxiv 2023.
>
> We hope that our rebuttal adequately addressed your concerns regarding ULSAD's novelty and performance. Thank you again for your reviews, which helped us improve our paper.

---

### Review · Reviewer_xveU · 2024-06-07

**Summary Of Contributions:**

The authors address the problem of industrial anomaly detection, which is challenging due to the diversity of anomaly types. The proposed method relies on Deep Feature Reconstruction (DFR) and is designed to detect logical and structural anomalies. The model uses a pre-trained backbone for extracting features and contains global and local loss elements to detect both types of anomalies. The authors have evaluated their method using several datasets and compared them to recently proposed schemes. The results presented in the paper support the strength of the model both in terms of image level and pixel level anomaly detection capabilities. They further conduct an ablation to support each component of their model.

**Audience:**

Yes

**Broader Impact Concerns:**

No concerns

**Claims And Evidence:**

Yes

**Requested Changes:**

On page 2, when you mention the pre-trained network \theta, it is worthwhile to mention what type of architecture this is.

Page 2, how are you selecting which layer $j$ to use?

Page 2, the function f is simply a reshape?

Based on section 3.2, I understand that the model is trained on clean samples. This type of training is often considered semi-supervised in AD. I believe this should be clarified early in the paper.

Following the previous comment, have the authors evaluated the influence of contaminated training set on the performance? Are all baselines trained the same way?

It isn’t clear how \tilde{U}’ and \tilde{U}’’ are trained. The first seems like a simple reconstruction loss, but I don’t see an explanation for the latter.

An evaluation of memory and compute requirements is missing, especially in comparison to existing works.

P11 “reverse distillation” is a name of a method? As such it should be capitalized.

**Strengths And Weaknesses:**

**Strengths**

The authors address an important and challenging problem in machine learning. The paper is overall well written and the English level is satisfactory.
The proposed method improves upon state-of-the-art models in terms of AUROC for image-level anomaly detection. Segmentation-level performance is also promising, leading to results comparable to “Reverse Distillation.”
The paper includes several complementary evaluations demonstrating each component's role in their model, such as hyperparameters, backbone models, and loss terms.



**Weaknesses**

The intuition behind some of the model components could be enhanced.
The proposed framework is based on several existing ideas. Nonetheless, I still consider the fusion of existing components into a new model a valid contribution to ML.

The authors mention the advantage of their model in terms of memory and efficiency of computation, but I don’t see any such comparison in the paper between their model and existing schemes.

---

> ### Author Response · Authors · 2024-08-08
> **Response to Reviewer xveU (1/2)**
>
> We thank the reviewer for their valuable feedback, suggestions and questions. We address the reviewer's concerns in the points below.
>
> **[W1]** The intuition behind some of the model components could be enhanced.
>
> Thank you for your constructive feedback. In our updated paper, we have improved the intuition in Sections 3.1, 3.2 and 3.3.
>
> **[W2, RC7]** The authors mention the advantage of their model in terms of memory and efficiency of computation. An evaluation of memory and compute requirements is missing.
>
> Please refer to our general comment on memory and computational complexity.
>
> **[RC1]** On page 2, when you mention the pre-trained network $\theta$, it is worthwhile to mention what type of architecture this is.
>
> In Section 3.1, we abstained from mentioning a specific choice for the pre-trained network $\theta$, as any deep neural network architecture used for representation learning can be used. Nonetheless, in the AD literature, CNN-based backbones, especially ResNets, are primarily used (Roth et al., 2022; Defard et al., 2021; Salehi et al., 2021). So, following the reviewer's suggestion and to aid the readers, we added a line in Section 3.1 mentioning the commonly used architecture for $\theta$. We additionally provide an ablation on the variant of ResNet used and the corresponding performance of ULSAD in Section 4.2.
>
> **[RC2]** Page 2, how are you selecting which layer $j$ to use?
>
> We follow PatchCore for extracting features from the pre-trained backbone. We extract the features from the second and third layers of a WideResNet50-2. We updated the implementation details in Section 4.1 to incorporate this detail.
>
> **[RC3]** Page 2, the function $f$ is simply a reshape?
>
> $f$ is indeed a reshape function as we mentioned in the last line of Section 3.1 in our paper. We wrote it as a separate function for ease of mathematical formulation.
>
> **[RC4]** Based on section 3.2, I understand that the model is trained on clean samples.
>
> We follow the existing literature for unsupervised AD (Roth et al., 2022; Bergmann et al., 2019), where the training data, although unlabeled, is assumed to be free from contamination (Ruff et al., 2021; Chandola et al., 2009). The goal is then to learn the features or representations that accurately capture ``normality". This setting is also referred to in the literature as one-class classification (Ruff et al., 2018). Moreover, the datasets used in our study contain only normal samples in the training split. Nonetheless, we acknowledge that such an assumption might be violated in practical scenarios, which has led to extensive research on unsupervised AD in the presence of contamination. However, we leave it for future study as it is beyond the scope of the current paper. We have mentioned the same in our updated draft as a limitation. Lastly, we do not refer to our method as semi-supervised (Ruff et al., 2020) as we do not have access to labelled samples during training.
>
> **[RC5]** Following the previous comment, have the authors evaluated the influence of contaminated training set on the performance? Are all baselines trained the same way?
>
> In this paper, we have not considered the impact of contamination in training the models, as handling contamination in itself is an active research topic. We will consider this in our future work. We trained all the baselines using the same training set as ULSAD. Moreover, to ensure a fair comparison, we used the Anomalib framework for all our experiments.
>
> **References:**
> - Roth et al., Towards Total Recall in Industrial Anomaly Detection, CVPR 2022.
> - Defard et al., PaDiM: a Patch Distribution Modeling Framework for Anomaly Detection and Localization, ICPR 2021.
> - Salehi et al., Multiresolution knowledge distillation for anomaly detection, CVPR 2021.
> - Bergmann et al., MVTEC ad-A comprehensive real-world dataset for unsupervised anomaly detection. CVPR 2019.
> - Ruff et al., A Unifying Review of Deep and Shallow Anomaly Detection, IEEE 2021.
> - Ruff et al., Deep One-Class Classification, ICML 2018.
> - Ruff et al., Deep semi-supervised anomaly detection, ICLR 2020.
> - Chandola et al., Anomaly detection: A Survey, ACM Computing Surveys, 2009.

---

> ### Author Response · Authors · 2024-08-08
> **Response to Reviewer xveU (2/2)**
>
> **[RC6]** It isn’t clear how $\widetilde{U}'$ and $\widetilde{U}''$ are trained.
>
> We train both $\widetilde{U}'$ and $\widetilde{U}''$ with a reconstruction loss which accounts for both the direction and value of the feature vectors. While for $\widetilde{U}'$ the $L_{pl}$ loss is computed between $\widetilde{U}'$ and the extracted feature map from the pre-trained network $U$, the $L_{lg}$ loss measures the difference between $\widetilde{U}''$ and the output from the global autoencoder $\widehat{U}$. The latter is used in the training of the global autoencoder to prevent false positives, as discussed in Section 3.3. Therefore, we deferred the details of $\widetilde{U}''$ to Section 3.3 Equation 6.
>
> **[RC8]** P11 “reverse distillation” is a name of a method?
>
> Thank you for pointing it out. We have updated this in our revised draft.
>
> We hope that our rebuttal adequately addressed your concerns regarding intuition behind model components, training setup and memory and computational complexity. Thank you again for your reviews, which helped us improve our paper.

---

### Review · Reviewer_msCY · 2024-07-26

**Summary Of Contributions:**

This paper proposes a new approach to industrial anomaly detection
(AD) capable of detecting both structural anomalies (local defects)
and logical anomalies (requiring global analysis of the product). The
approach extends an existing method, deep feature reconstruction
(DFR), which uses features extracted by a pretrained network, to
improve identification of anomalies. Extensions include a modification
of the loss to include the angular distance between network outputs
and targets as well as a larger extension adding a second sub-network
for global analysis to detect logical anomalies. The authors test the
new approach on a variety of AD datasets and compare against existing
approaches as a baseline, and perform ablation tests to measure the
improvements provided by the various components of the overall method.

**Audience:**

Yes

**Broader Impact Concerns:**

No concerns

**Claims And Evidence:**

Yes

**Requested Changes:**

A couple things that I think would provide a nice improvement if they
are feasible, then a few questions and minor notes. Overall, I
appreciate the thorough ablation and tuning results.

*[Important]* I think it would help contextualize performance if you
are able to provide some estimate or discussion of the cost of ULSAD
vs. some of the other approaches (in particular vs. nearest-neighbor
methods like PatchCore). You mention that the lack of a memory bank is
an advantage which could compensate for slightly lower performance in
some cases. It would be nice to have some idea of how much of an
advantage this is, if possible (i.e. how much memory is saved by
losing the memory bank, or perhaps inference time improvements,
something along those lines.). At least a rough idea to give readers
some context as to the costs of the memory or nearest neighbor search
on some of these tasks.

*[Moderate]* It would be interesting to see the MVTec-Loco performance
split between logical/structural errors if those results are
available, to compare performance against baselines between anomaly
types. I understand, though, if this would not be possible without
excessive computational expense.

*[Question]* among your baselines it appears that EfficientAD is the
only network that ran tests against MVTec-Loco, is that right?

*[Question]* On the performance of EfficientAD vs. the source paper:
the original EfficientAD paper (Batzner, et al 2024) appears to report
higher anomaly detection performance on MVTec-Loco (roughly 90.0
averaged across tasks) while your experiments hover closer to 80. Do
you know if you made any changes to the baseline that might account
for some of these differences?

The Batzner paper seems to indicate it uses AU-sPRO for the Loco
results, which may explain a difference for the segmentation; perhaps
there is something similar for the detection results. I don't doubt
that these are the results you obtained from testing, I am just
curious if you have thoughts on what might have produced the
differences.

*[Note]* the list of baselines in the body text under Table 2 is
missing EfficientAD.

*[Note]* Are the tables of results in the appendices intended to have
1st and 2nd place results bolded/underlined? This seems to only be
true for some rows and tables.

**Strengths And Weaknesses:**

**Strengths**
- Clearer performance improvements on more challenging cases (i.e.
  MVTec-Loco including logical defects)
- Generally clear exposition detailing the approach and the intended
  purpose of each component and extension
- Ablation tests to support the contribution of each component to
  overall improvement
- Code available and implemented in an AD framework which should
  permit others to compare against this approach
- Appendix provides relatively detailed performance information
  (per-task in each dataset)

**Weaknesses**
- Relative performance on simpler structural-defect-only datasets
  (those other than MVTec-Loco) is somewhat harder to judge,
  improvements here, if any, are smaller
- Could use additional discussion of trends in the results tables to
  help readers understand them
- Some measures appear to differ vs baseline works which tested on the
  MVTec-Loco dataset. Comparisons here are self-consistent but this
  may make it harder to compare results against other papers on that
  dataset
- Approach appears to be tuned (for ex. selection of parameters
  lambda) on MVTec-Loco, which may contribute to reduced performance
  on the other datasets

---

> ### Author Response · Authors · 2024-08-08
> **Response to Reviewer msCY (1/2)**
>
> We thank the reviewer for their constructive feedback. We address the reviewer's concerns in the points below.
>
> **[W1]** Relative performance on simpler structural-defect-only datasets is somewhat harder to judge.
>
> We acknowledge that the performance of ULSAD on simpler structural-defect-only datasets is similar to the SOTA methods. However, our goal in this paper is to improve DFR, which we identified as an efficient method both computationally and in terms of memory requirement. Our experiments demonstrate that the incorporation of recent advancements in AD literature improves the performance of DFR to deliver competitive results when compared to SOTA. Furthermore, ULSAD can detect logical anomalies, a capability that DFR lacks.
>
> **[W2]** Could use additional discussion of trends in the results tables.
>
> We have incorporated an additional discussion of trends in Section 4.2 of our updated paper. We additionally highlight some key points below:
>
> In Table 1, we showcase the anomaly detection performance of ULSAD on 5 benchmarking datasets and compare it with other methods in the literature. We observe that in BTAD, MPDD, VisA and MVTecLOCO, ULSAD has a better performance than the rest including DFR. On MVTec, the performance is similar to that of PatchCore, which is a memory-bank-based approach. Therefore, it can be said that on MVTec, ULSAD offers similar performance while having substantially less memory requirement than PatchCore.
>
> Overall, ULSAD demonstrates competitive results in anomaly detection compared to the baseline methods across all benchmark datasets. Additionally, the difference in performance between ULSAD and the baselines for anomaly localization is minimal. The most notable difference is in the AUPRO score on the BTAD, MVTec, and VisA datasets. Nonetheless, while the SOTA method provides slightly better performance in the localization of structural anomalies, ULSAD provides similar performance across both logical and structural anomalies.
>
> Additionally, we provide results for MVTecLOCO, focusing on structural and logical anomalies separately in Appendix B.1.
>
> **[W3]** Some measures appear to differ vs baseline works which tested on the MVTec-Loco dataset.
>
> To evaluate the anomaly localization performance across various IAD datasets, existing works use either PRO (Roth et al., 2022), a threshold-dependent metric, or AU-PRO (Bergmann et al., 2019), which is independent of threshold selection, for anomaly localization. Both metrics measure the overlap between the predicted area and the ground truth annotation. AU-sPRO (Bergmann et al., 2022) is a generalization of the AU-PRO, where a saturation threshold is defined for each category. The segmentation task is considered solved when the overlap between the predicted anomalous area and the ground truth exceeds the saturation threshold. However, the saturation thresholds are only available for logical anomalies in the MVTecLOCO dataset. Therefore, to maintain consistency across benchmark datasets, following Bergmann et al., 2019 and Batzner et al., 2024, we report the AU-PRO scores up to a false positive rate of $30\\%$.
>
> **[W4]** Approach appears to be tuned on MVTec-Loco.
>
> For selecting the hyperparameters, we considered all the benchmark datasets and chose the ones that provide similar performance across all the datasets for anomaly detection and localization.
>
> **[RC1]** I think it would help contextualize performance if you are able to provide some estimate or discussion of the cost of ULSAD vs some of the other approaches.
>
> Please refer to our general comment on memory and computational complexity.
>
> **[RC2]** It would be interesting to see the MVTec-Loco performance split between logical/structural errors.
>
> Following the reviewer's comment, we report the anomaly detection and localization results on MVTec LOCO separately for structural and logical anomalies in Appendix B.1 of our updated draft. We observe that although PatchCore performs slightly better than ULSAD on structural anomalies, ULSAD delivers competitive results on logical anomalies. Moreover, as discussed in Section 5 of our updated draft, the improvement by 1-2 points in performance comes with three times the memory requirement when using PatchCore. Thus, we consider ULSAD to be an efficient and effective approach for the detection and localization of both logical and structural anomalies.
>
> **References:**
> - Roth et al., Towards Total Recall in Industrial Anomaly Detection, CVPR 2022.
> - Bergmann et al., MVTEC ad-A comprehensive real-world dataset for unsupervised anomaly detection. CVPR 2019.
> - Batzner et al., EfficientAD: Accurate Visual Anomaly Detection at Millisecond-Level Latencies, WACV 2024.
> - Bergmann et al., Beyond Dents and Scratches: Logical Constraints in Unsupervised Anomaly Detection and Localization, IJCV, 2022

---

> ### Author Response · Authors · 2024-08-08
> **Response to Reviewer msCY (2/2)**
>
> **[RC3]** Among your baselines it appears that EfficientAD is the only network that ran tests against MVTec-Loco, is that right?
>
> Yes. Among the baselines, only EfficientAD was tested on MVTec LOCO. We chose this method as it is the current SOTA method for detection of logical anomalies.
>
> **[RC4]** On the performance of EfficientAD vs. the source paper: the original EfficientAD paper (Batzner, et al 2024) appears to report higher anomaly detection performance on MVTec-Loco.
>
> We acknowledge the difference in results for EfficientAD between the original paper (Batzner et al., 2024) and our experiments. For the detection results, we use the same metric, AUROC, as the authors of EfficientAD. As the original code is not released by the authors we used the implementation in the Anomalib library for our experiments. We spent a substantial amount of time trying to validate the code and tune the parameters. However, despite our best efforts, we could not replicate the performance as reported. Moreover, we also contacted the authors for the code but were informed that the code is confidential and cannot be shared. Therefore, we resorted to reporting the results we obtained. Furthermore, to ensure a fair comparison, we also used the Anomalib framework for evaluating ULSAD.
>
> **[RC4]** The list of baselines in the body text under Table 2 is missing EfficientAD.
>
> Thank you for bringing this to our attention. We have added EfficientAD to the list of baselines.
>
> **[RC5]** Are the tables of results in the appendices intended to have 1st and 2nd place results bolded/underlined?
>
> Thank you for pointing out this out. We have fixed the format in our updated draft.
>
> **References:**
> - Batzner et al., EfficientAD: Accurate Visual Anomaly Detection at Millisecond-Level Latencies, WACV 2024.
>
> We hope that our rebuttal adequately addressed your concerns regarding the results, evaluation metrics, memory, and computational complexity. Thank you again for your reviews, which helped us improve our paper.

---

> > ### Comment · Reviewer_msCY · 2024-08-28
> >
> > Thank you for your thorough response to my questions. I especially appreciate your willingness to extend your results to distinguish split between logical/structural anomalies.
> >
> > One quick note, looking over your revisions to the paper it looks like in the new paragraph for section 6, the word `"clean"` uses plain straight quotes, you might want to consider LaTeX open and close quotes (i.e. ``` ``clean'' ```) which may look slightly better.

---

### Author Response · Authors · 2024-08-08
**General Comment on Memory and Computational Complexity**

- **[Reviewer msCY]** I think it would help contextualize performance if you are able to provide some estimate or discussion of the cost of ULSAD vs some of the other approaches.
- **[Reviewer xveU]** The authors mention the advantage of their model in terms of memory and efficiency of computation. An evaluation of memory and compute requirements is missing.
- **[Reviewer UCFP]** Compare the model size and computational resources.

ULSAD is computationally inexpensive as it operates on the lower dimensional latent space of a deep feature reconstruction network instead of operating on the high dimensional image space. Following the reviewers' comments, we have compared the throughput and GPU memory usage of ULSAD against the baselines in Section 5 of our updated draft. We can observe that ULSAD has a higher throughput than most of the baselines while demonstrating competitive anomaly detection and localization performance. Moreover, ULSAD requires $ \sim 1/3 $rd the memory when compared to retrieval-based methods such as PatchCore, which is one of the SOTA methods for industrial anomaly detection.

---

### Decision · Action_Editor_aQLe · 2024-08-30

**Recommendation:** Accept as is

**Comment:**

The authors have addressed all major concerns raised in the reviews. It is agreed that the paper is ready for publication.

**Audience:**

The paper would be of interest to the those who work on anomaly detection and some of those who work on ML applications.

**Claims And Evidence:**

The paper aims to develop solutions to logical and structural industrial anomaly detection, which is an important problem in ML applications. The proposed solution is built based on deep Feature Reconstruction (DFR), which is both memory- and compute-efficient.  Experiment results have demonstrated its effectiveness. In general, the claims in the submission are well-supported.